# Cerebellar-Inspired Residual Control for Fault Recovery: From Inference-Time Adaptation to Structural Consolidation

**Nethmi Jayasinghe** [1]   **Diana Gontero** [1]   **Spencer T. Brown** [2]   **Vinod K. Sangwan** [3]   **Mark C. Hersam** [3][4][5]
**Amit Ranjan Trivedi** [1]

## Abstract

Robotic policies deployed in real-world environments often encounter post-training faults, where retraining, exploration, or system identification are impractical. We introduce an inference-time, cerebellar-inspired residual control framework that augments a frozen reinforcement learning policy with online corrective actions, enabling fault recovery without modifying base policy parameters. The framework instantiates core cerebellar principles, including high-dimensional pattern separation via fixed feature expansion, parallel microzone-style residual pathways, and local error-driven plasticity with excitatory and inhibitory eligibility traces operating at distinct time scales. These mechanisms enable fast, localized correction under post-training disturbances while avoiding destabilizing global policy updates. A conservative, performance-driven meta-adaptation regulates residual gain and plasticity, preserving nominal behavior and suppressing unnecessary intervention. Experiments on MuJoCo benchmarks under actuator, dynamic, and environmental perturbations show improvements of up to +66% on `HalfCheetah-v5` and +53% on `Humanoid-v5` under moderate faults, with graceful degradation under severe shifts and complementary robustness from consolidating persistent residual corrections into policy parameters.

[1]Department of Electrical and Computer Engineering, University of Illinois Chicago, USA [2]Department of Neurobiology, Northwestern University, USA [3]Department of Materials Science and Engineering, Northwestern University, USA [4]Department of Electrical and Computer Engineering, Northwestern University, USA [5]Department of Chemistry, Northwestern University, USA. Correspondence to: Nethmi Jayasinghe <wjayas3@uic.edu>, Amit Ranjan Trivedi <amitrt@uic.edu>.

*Proceedings of the 43rd International Conference on Machine Learning*, Seoul, South Korea. PMLR 306, 2026. Copyright 2026 by the author(s).

## 1. Introduction

Humans adapt to unexpected physical disturbances rapidly and safely, without explicit retraining or exploration. For example, when lifting an object that is lighter or heavier than anticipated, the nervous system applies corrective forces within a single movement, preventing loss of balance or grip. This behavior reflects learning mechanisms that operate continuously at inference time, detecting deviations from prediction, applying fast local correction, and suppressing unnecessary intervention when behavior remains nominal.

In contrast, modern robotic control policies are typically trained offline under fixed assumptions about system dynamics and disturbances. Once deployed, these policies lack mechanisms for safe and rapid adaptation to unforeseen post-training faults, such as actuator degradation, unmodeled dynamics shifts, or changes in environmental contact conditions. Retraining, exploration, or system identification at deployment is often infeasible due to safety constraints, limited data, or computational cost, leading to substantial performance degradation or instability.

This mismatch highlights a gap between how humans and robots adapt. Human motor control relies heavily on the cerebellum, a structure specialized for high-throughput sensory processing, prediction error detection, and fast, local error-driven learning. Despite dense sensory input, the cerebellum suppresses predicted signals, detects unpredicted deviations, and applies rapid corrective actions without altering the underlying motor plan. These properties are well suited for handling transient disturbances while preserving stable nominal behavior.

Motivated by this observation, we propose an inference-time, cerebellar-inspired residual control architecture that augments a *frozen* reinforcement learning policy with lightweight online correction. The approach decouples long-horizon policy learning from fast adaptation: the base policy encodes nominal task behavior and remains unchanged at deployment, while a parallel residual pathway provides rapid correction under post-training disturbances. This residual pathway instantiates core cerebellar computational principles in algorithmic form, including high-dimensional pattern

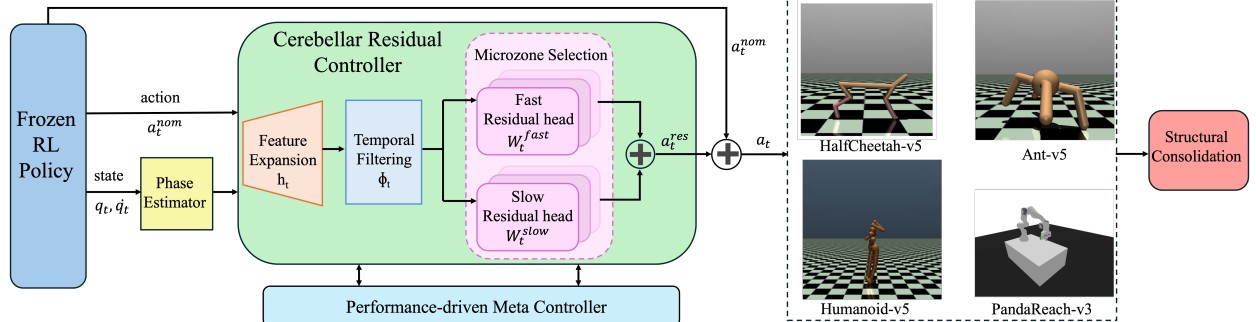

*Figure 1.* Overview of the cerebellar-inspired fault recovery framework. A frozen base policy is augmented with an inference-time residual controller that learns local, error-driven corrections under post-training faults, enabling rapid recovery without modifying the base policy. Under persistent faults, residual structure is consolidated into a static adapter to maintain long-term robustness. The framework is evaluated on locomotion tasks (`HalfCheetah-v5`, `Ant-v5`, `Humanoid-v5`) and a non-cyclic manipulation task (`PandaReach-v3`), demonstrating scalability across control dimensionality and task structure.

separation via fixed feature expansion, parallel microzone-style pathways for localized correction, and excitatory and inhibitory eligibility traces operating at distinct time scales for temporal credit assignment. Residual corrections are learned from local tracking errors and applied additively, preserving the structure of the learned policy, while a performance-driven meta-adaptation mechanism regulates residual gain and plasticity based on sustained reward trends, ensuring intervention only under persistent degradation.

From a learning perspective, the proposed approach operates in a complementary regime that requires neither fault exposure during training nor exploration, optimization, or system models at deployment, instead combining the representational capacity of deep reinforcement learning with the efficiency and stability of cerebellar-style error-driven learning to enable safe inference-time adaptation under post-training faults. Across MuJoCo locomotion benchmarks spanning increasing control complexity from `HalfCheetah-v5` to `Humanoid-v5`, inference-time residual adaptation improves episodic return by approximately +66% and +53% respectively under moderate faults, while degrading gracefully under more severe whole-body perturbations.

## 2. Related Work

**Training-Time Robustness.** A common approach to handling disturbances in robotic control is to improve robustness during training through domain randomization (Tobin et al., 2017; Peng et al., 2018) or robust reinforcement learning (Pinto et al., 2017; Rajeswaran et al., 2016). These methods optimize policies to perform well on average over predefined distributions of dynamics variations, requiring prior specification of fault families and severity ranges. While effective against anticipated disturbances, such approaches often degrade under out-of-distribution post-training faults and provide no mechanism for recovery when failures arise after deployment. In contrast, our work targets fault recovery *after* training, without relying on fault exposure during optimization or modifying the learned policy.

**System Identification and Dynamics-Conditioned Policies.** Post-training adaptation has also been studied through online system identification or dynamics-conditioned policies (Yu et al., 2017; Nagabandi et al., 2018; Kumar et al., 2021). These methods infer latent dynamics parameters and condition control actions on the estimated state of the system, enabling adaptation when faults admit a low-dimensional, identifiable structure that is well covered during training. However, such assumptions often break down in contact-rich locomotion, where faults may be localized, transient, or poorly parameterized. Our approach avoids explicit system identification altogether and adapts directly from online tracking and performance signals, enabling recovery even when fault is unknown or difficult to model.

**Residual Learning and Adaptive Control.** Residual learning augments nominal controllers or policies with corrective actions, either through offline training under known disturbances (Johannink et al., 2019; Silver et al., 2018) or through online adaptation using local error signals (Hovakimyan & Cao, 2010). Classical adaptive control offers strong theoretical guarantees for structured uncertainty (Slotine & Li, 1991), but typically relies on explicit system models or low-dimensional uncertainty parameterizations. Naive online residual adaptation, however, often suffers from instability or drift, as learning proceeds continuously regardless of task-level performance. Our method draws inspiration from adaptive control while departing in two key respects: it operates without explicit models or policy-gradient updates, and it regulates residual plasticity and gain conservatively using sustained performance signals, preserving nominal behavior by default.

**Cerebellar-Inspired Learning.** The cerebellum has long been studied as a substrate for fast, local, error-driven motor adaptation, formalized by Feedback Error Learning and

related models (Albus, 1971; Kawato & Gomi, 1992). Prior cerebellar-inspired controllers have demonstrated effective adaptation in structured or low-dimensional settings (Nakanishi & Schaal, 2004), but their integration with deep reinforcement learning and high-dimensional locomotion remains limited. Our work abstracts core cerebellar computational principles, high-dimensional pattern separation, parallel microzone organization, and multi-timescale error-driven plasticity, into an inference-time residual architecture compatible with modern deep RL policies.

**Inference-Time Adaptation.** Recent work in machine learning has explored test-time or inference-time adaptation to distribution shifts after training, primarily in supervised or representation learning settings. Such methods typically rely on parameter updates at test time driven by unlabeled data or surrogate objectives, such as self-supervised losses or entropy minimization (Sun et al., 2020; Wang et al., 2021). They also assume open-loop inference without safety-critical feedback. In contrast, inference-time adaptation in control must operate under closed-loop dynamics, avoid destabilizing exploration, and preserve nominal behavior. Prior work on safe online learning and adaptive control has emphasized the difficulty of guaranteeing stability and bounded behavior under such conditions, especially in high-dimensional nonlinear systems (Berkenkamp et al., 2017; Dean et al., 2020). Our approach addresses this regime by enabling conservative, performance-gated adaptation at inference time without updating the base policy. Overall, existing approaches to robustness and post-training adaptation in control rely on training-time disturbance exposure, explicit dynamics inference, or retraining. The proposed approach occupies a complementary regime, enabling fault-agnostic, conservative inference-time recovery by combining the representational capacity of deep reinforcement learning with cerebellar-style error-driven adaptation while preserving nominal behavior.

## 3. Inference-Time Residual Adaptation and Learning Consolidation

Robotic control policies trained offline encode long-horizon task competence under nominal assumptions about dynamics and actuation. After deployment, unknown post-training faults, such as actuator degradation or unmodeled dynamics shifts, can violate these assumptions. In safety-critical settings, retraining, exploration, or explicit system identification at deployment is often infeasible. We study how a frozen policy can be augmented at inference time to recover from such faults while preserving nominal behavior and closed-loop stability, and how fast corrective adaptation can be periodically consolidated to regenerate capacity for continual operation over the system lifetime. A unified algorithmic summary of the inference-time residual adaptation

procedure is provided in Appendix A.

**Deployment Constraints and Design Requirements.** We consider a continuous-control MDP in which a base policy $\pi_\theta : \mathcal{S} \to \mathcal{A}$ is trained offline in a nominal, fault-free environment and then frozen. At deployment, unknown and potentially transient faults may alter the system dynamics. Deployment is subject to strict constraints: policy parameters remain fixed, no exploration or policy-gradient optimization is permitted, no explicit system identification or fault supervision is available, and all adaptation must occur online and in closed loop. The objective is to preserve nominal behavior in the absence of faults and recover performance under faults while ensuring bounded behavior.

Under these constraints, any viable inference-time adaptation mechanism must satisfy the following requirements: (i) **Additivity**, corrective actions are applied additively to the nominal policy; (ii) **Locality**, adaptation relies on short-horizon error signals rather than delayed rewards; (iii) **Bounded Authority**, corrective influence is explicitly regulated; (iv) **Timescale Separation**, fast disturbance correction is decoupled from long-horizon learning; (v) **Default Inactivity**, adaptation remains inactive in the absence of sustained degradation; and (vi) **Capacity Regeneration**, persistent corrections are transferred to a slower learner to prevent saturation of the fast adaptive pathway.

These requirements rule out direct policy adaptation, inference-time reward optimization, and classical adaptive control as standalone solutions under unknown, high-dimensional faults, and instead identify a residual adaptive corrector layered on top of a stable controller as a complementary and minimal structure for safe inference-time recovery and continual operation.

**Control Decomposition and Cerebellar Insight.** We adopt a composite control formulation

$$a_t = a_t^{\text{nom}} + a_t^{\text{res}}, \tag{1}$$

where $a_t^{\text{nom}} = \pi_\theta(s_t)$ is the action produced by the frozen policy and $a_t^{\text{res}}$ is an adaptive residual correction. This decomposition enforces a separation of responsibilities: the nominal policy encodes long-horizon task structure and stability, while the residual pathway provides fast, local compensation for post-training disturbances without modifying the base policy.

Inference-time adaptation is framed not as optimization but as *regulation of corrective authority*. Corrective signals must be bounded, assistive, and suppressible, rather than globally optimizing behavior. This aligns with the functional role of the cerebellum in motor control, which operates as an adaptive corrector layered on top of a stable controller, with fast error-driven correction and slow consolidation of persistent corrections into long-term structure. We adopt this

abstraction explicitly: the residual pathway adapts rapidly to deviations, while persistent corrections are consolidated into the base policy over longer timescales.

**Phase-Aligned Reference and Localized Residual Structure.** Residual adaptation is driven by deviations relative to a nominal reference trajectory generated offline by rolling out the frozen policy in a fault-free environment. The reference is indexed by a normalized phase variable $\phi_t \in [0, 1)$ rather than absolute time. During deployment, $\phi_t$ is estimated online from joint kinematics, enabling phase-aligned reference lookup without gait scheduling, replanning, or trajectory regeneration.

To enable localized, context-dependent adaptation, the residual pathway is partitioned into phase-conditioned microzones. At each time step, the estimated phase $\phi_t$ activates a small subset of microzones through smooth interpolation. This structure restricts learning to phase-relevant contexts, prevents global interference with the base policy, and mirrors cerebellar microzone organization.

**Residual Parameterization and Multi-Timescale Adaptation.** At each time step, the residual controller receives joint-level kinematic signals

$$x_t = [q_t, \ \dot{q}_t, \ \ddot{q}_d(\phi_t)], \tag{2}$$

where $q_t$ and $\dot{q}_t$ are measured joint positions and velocities, and $\ddot{q}_d(\phi_t)$ is the phase-indexed desired joint acceleration from the nominal reference. The input is projected into a high-dimensional feature via a fixed random matrix $V$,

$$h_t = \sigma(V x_t), \tag{3}$$

and filtered using dual exponential traces

$$\Phi_t^E = \Phi_{t-1}^E + \alpha_E(h_t - \Phi_{t-1}^E), \tag{4}$$

$$\Phi_t^I = \Phi_{t-1}^I + \alpha_I(h_t - \Phi_{t-1}^I), \tag{5}$$

with $\alpha_E > \alpha_I$. The effective activity $\Phi_t = \Phi_t^E - \Phi_t^I$ provides short- and long-timescale eligibility signals without online representation learning. The residual controller maintains separate fast and slow weight matrices within each microzone. The residual action is computed as

$$a_t^{\mathrm{res}} = g_t \left( W_t^{\mathrm{fast}} + W_t^{\mathrm{slow}} \right) \Phi_t, \tag{6}$$

where $g_t \in [0, g_{\max}]$ regulates residual gain. Fast weights adapt rapidly to transient disturbances, while slow weights accumulate persistent, phase-specific corrections suitable for consolidation. A directional consistency gate suppresses residual actions that oppose the nominal policy output, ensuring that residual adaptation remains assistive and inactive under nominal conditions.

**Error-Driven Local Learning.** Residual adaptation is driven by joint-level tracking error relative to the phase-aligned reference,

$$e_t = q_d(\phi_t) - q_t, \qquad \dot{e}_t = \dot{q}_d(\phi_t) - \dot{q}_t, \tag{7}$$

from which we define a composite error

$$r_t = \dot{e}_t + \Lambda e_t, \tag{8}$$

where $\Lambda$ is a diagonal gain matrix. For the active microzone, fast and slow residual weights are updated using a normalized local adaptive rule

$$\Delta W_t^{(\cdot)} = \eta^{(\cdot)} \frac{r_t \Phi_t^\top}{\|\Phi_t\|_2 + \epsilon}, \tag{9}$$

with deadzones and regularization to suppress drift when errors are small. All learning is local, linear in parameters, and confined to the active microzone.

**Meta-Controller and Authority Regulation.** Residual adaptation is regulated by a meta-controller that modulates corrective authority based on sustained performance trends rather than instantaneous reward. Let $\bar{R}_t$ denote an exponential moving average of task reward,

$$\bar{R}_t = (1 - \rho)\bar{R}_{t-1} + \rho R_t. \tag{10}$$

The meta-controller adjusts residual gain according to bounded relaxation dynamics,

$$g_{t+1} = g_t + \kappa \, \mathbb{I}[\bar{R}_t < \bar{R}^\star - \delta] - \lambda g_t, \tag{11}$$

where $\bar{R}^\star$ is the best observed performance. This mechanism treats the meta-controller as an authority regulator rather than an optimizer, enforcing default inactivity and preserving nominal behavior.

**Capacity Regeneration through Consolidation.** Because the residual pathway has finite adaptive capacity, persistent nonzero residual activity indicates a systematic mismatch between the frozen policy and deployment dynamics. Let the episodic residual energy be

$$E_{\mathrm{res}} = \frac{1}{T} \sum_{t=1}^{T} \|a_t^{\mathrm{res}}\|^2. \tag{12}$$

Residual energy serves as a diagnostic for persistent corrective effort during consolidation, while online adaptation is activated by the reward-EMA meta-controller described above. Sustained elevation of $E_{\mathrm{res}}$ signals structural bias rather than transient disturbance. In this regime, trajectories generated under residual-assisted execution are used to consolidate the slow residual weights into an updated base policy or static linear adapter via offline regression. Consolidation is successful when

$$\mathbb{E}_{\pi_{\theta'}}[E_{\mathrm{res}}] \ll \mathbb{E}_{\pi_\theta}[E_{\mathrm{res}}], \tag{13}$$

returning the residual pathway to its default inactive state and restoring adaptive capacity for future faults. This two-timescale interaction mirrors cerebellar–cortical learning, enabling continual fault recovery without saturation or catastrophic interference.

# 4. Safety and Stability Properties

Rather than pursuing global convergence or optimality guarantees, which are intractable for high-dimensional locomotion under unknown faults, we formalize properties that follow directly from architectural constraints: bounded corrective authority, non-interference under nominal conditions, phase-local adaptation, and controlled adaptation with capacity regeneration.

**Bounded Residual Gain.** The executed control is

$$a_t = \pi_\theta(s_t) + a_t^{\mathrm{res}}, \qquad a_t^{\mathrm{res}} = g_t \left( W_{k_t}^{\mathrm{fast}} + W_{k_t}^{\mathrm{slow}} \right) \Phi_t, \tag{14}$$

where $k_t$ denotes the active phase-conditioned microzone. By construction, residual features satisfy $\|\Phi_t\|_2 \leq \Phi_{\max}$, residual gain is bounded as $0 \leq g_t \leq g_{\max}$, and adaptive weights are projected onto a compact set $\|W\|_F \leq W_{\max}$. These constraints imply a uniform bound

$$\|a_t^{\mathrm{res}}\|_2 \leq g_{\max} W_{\max} \Phi_{\max} \triangleq a_{\max}, \tag{15}$$

independent of the fault realization. As a result, inference-time adaptation cannot inject unbounded control inputs.

**Non-Interference Under Nominal Conditions.** Residual gain is regulated rather than optimized. In the absence of sustained performance degradation, the meta-controller enforces dissipative gain dynamics

$$g_{t+1} = (1 - \lambda)g_t, \qquad \lambda > 0, \tag{16}$$

implying $g_t \to 0$ and hence $a_t^{\mathrm{res}} \to 0$. The closed-loop behavior therefore asymptotically recovers that of the frozen policy $\pi_\theta$. In addition, residual actions are restricted to be assistive, with corrections opposing the base policy suppressed. Together, these mechanisms ensure structural non-interference under nominal conditions.

**Phase-Local Adaptation and Interference Containment.** Residual learning and actuation are confined to the active phase-conditioned microzone determined by the online phase estimate. Adaptive updates and corrective actions therefore remain localized to the current phase of motion, preventing errors or faults in one phase from propagating to others. This phase locality bounds the spatial and temporal footprint of inference-time adaptation and limits unintended cross-context interference.

**Controlled Adaptation and Capacity Regeneration.** When post-training faults induce sustained tracking error, residual adaptation is activated through bounded gain modulation and local error-driven learning. Because the residual pathway operates additively and leaves the base policy unchanged, persistent residual activity indicates a systematic mismatch between the frozen policy and deployment dynamics rather than transient disturbance. Using the episodic

residual energy $E_{\mathrm{res}}$ defined in Section 3, sustained elevation of $E_{\mathrm{res}}$ signals structural bias that cannot be eliminated by transient correction alone. In this regime, residual-assisted trajectories are consolidated into the base policy or a static linear adapter through offline or episodic fine-tuning, yielding an updated policy $\pi_{\theta'}$ such that

$$\mathbb{E}_{\pi_{\theta'}}[E_{\mathrm{res}}] \ll \mathbb{E}_{\pi_\theta}[E_{\mathrm{res}}]. \tag{17}$$

This two-timescale structure, combining fast bounded correction with slow structural learning, enables continual post-training adaptation while preserving stability and preventing catastrophic interference.

# 5. Experimental Setup

We evaluate the proposed inference-time cerebellar residual controller on continuous-control locomotion tasks under post-training faults, testing whether performance can be recovered at deployment without retraining or modifying the base policy.

**Environments.** Experiments are conducted in MuJoCo benchmarks using the Gymnasium interface. We evaluate locomotion tasks with increasing control complexity, including `HalfCheetah-v5`, `Ant-v5`, and `Humanoid-v5`, spanning moderate to high coordination complexity. To assess generality beyond periodic locomotion, we additionally evaluate a non-cyclic manipulation task, `PandaReach-v3`, using the `panda_gym` suite. All environments use default observation and action spaces, control frequencies, termination conditions, and episode horizons. Evaluation rollouts use the deterministic policy (mean action) without exploration noise.

**Base Policies and Reference Trajectories.** Base policies are trained under nominal, fault-free dynamics using Soft Actor-Critic (SAC). After training, policies are frozen and evaluated without policy-gradient updates, replay buffers, or parameter changes. Nominal reference trajectories used by the residual controller are generated by rolling out the frozen policy in the fault-free environment and recording joint positions and velocities; desired accelerations are obtained via finite differences and indexed by timestep during deployment.

**Fault Models.** Faults are introduced only at evaluation time (deployment) after policy training and are persistent, unlabeled, and unknown to the controller within each rollout. We evaluate three fault families: (i) *actuator faults*, including multiplicative actuator scaling applied uniformly across joints (scaling factors in $[0.4, 0.9]$) and additive actuator bias applied as constant offsets; (ii) *dynamic faults*, including increased body mass (multipliers in $[1.2, 1.6]$) and increased joint damping (multipliers in $[1.2, 2.2]$); and (iii) *environmental faults*, including decreased friction to model slippery

*Table 1.* Performance under post-training faults (mean ± std episodic return over 30 runs). One representative moderate fault severity is reported per fault family. Robust SAC, OSI, and Offline Res. utilize training-time fault exposure, whereas CMAC, LMS, and Ours adapt online at inference time. **Bold** indicates best; underlined indicates second-best.

| Category | Method | HalfCheetah-v5 | | | Ant-v5 | | |
|---|---|---|---|---|---|---|---|
| | | Actuator | Dynamic | Environment | Actuator | Dynamic | Environment |
| Classic RL | SAC | $3612 \pm 594$ | $5093 \pm 1165$ | $11958 \pm 4085$ | $4542 \pm 907$ | $3816 \pm 1909$ | $3796 \pm 1397$ |
| Train with faults | Robust SAC | $3093 \pm 626$ | $3491 \pm 541$ | $12315 \pm 410$ | $5278 \pm 180$ | $4001 \pm 1203$ | $3528 \pm 1594$ |
| Fault-aware | SAC + OSI | $\mathbf{7548 \pm 250}$ | $\mathbf{8810 \pm 251}$ | $11738 \pm 2350$ | $3128 \pm 711$ | $3376 \pm 1223$ | $2207 \pm 806$ |
| Residual | SAC + Offline Res. | $3898 \pm 200$ | $5441 \pm 824$ | $12977 \pm 1573$ | $5220 \pm 1300$ | $\mathbf{4918 \pm 1734}$ | $3323 \pm 1840$ |
| Residual | SAC + CMAC | $3855 \pm 774$ | $6052 \pm 949$ | $11011 \pm 4821$ | $5324 \pm 2403$ | $4556 \pm 1265$ | $\mathbf{4985 \pm 2019}$ |
| Residual | SAC + Online LMS | $4000 \pm 782$ | $5517 \pm 1140$ | $11672 \pm 4184$ | $4273 \pm 1138$ | $4712 \pm 2176$ | $2984 \pm 499$ |
| Residual | SAC + Ours | $7239 \pm 2159$ | $7502 \pm 1281$ | $\mathbf{19517 \pm 3834}$ | $\mathbf{5622 \pm 1137}$ | $4728 \pm 1447$ | $4551 \pm 2132$ |

contact and increased friction to model high-traction terrain (scaling factors in $[0.1, 2.0]$). Each fault type is evaluated across multiple severity levels. For tabular reporting, we use one representative moderate-severity fault per family, namely actuator scaling, damping and friction increase.

**Baselines.** We compare against baselines spanning non-adaptive policies, training-time robustness, fault-aware policies, and residual learning methods. Robust SAC represents training-time robustness through domain randomization and robust RL (Tobin et al., 2017; Peng et al., 2018; Pinto et al., 2017; Rajeswaran et al., 2016). SAC+OSI represents dynamics-conditioned adaptation with online system identification (Yu et al., 2017; Kumar et al., 2021). Residual baselines include (i) offline-trained residuals exposed to faults during training (Johannink et al., 2019; Silver et al., 2018) and (ii) inference-time adaptive residual methods such as online LMS and CMAC (Hovakimyan & Cao, 2010; Slotine & Li, 1991). The frozen SAC policy is included to quantify fault-induced degradation under post-training shifts, rather than as the primary competing method. All methods are evaluated under identical fault settings and budgets. Performance is measured by cumulative episodic return over a fixed evaluation horizon after fault injection. Each fault setting is evaluated over 10 random seeds with 3 episodes per seed (30 rollouts per condition). We report the mean and standard deviation of episodic return across rollouts.

## 6. Results and Discussions

**Generalization Across Fault Types and Severities.** We evaluate generalization across fault types and severity levels on HalfCheetah-v5. Figure 3 reports episodic returns under a broad set of actuator, dynamic, and environmental perturbations spanning multiple fault families and magnitudes. Across all evaluated fault types and severity levels, the proposed controller consistently improves performance over the faulted frozen-policy baseline. Inference-time residual adaptation remains effective throughout the severity sweep, demonstrating robust recovery across a wide range

*Table 2.* *Humanoid-v5* performance under post-training faults (mean ± std episodic return over 30 runs). One representative moderate fault severity is reported per fault family. **Bold** indicates best; underlined indicates second-best.

| Method | Humanoid-v5 | | |
|---|---|---|---|
| | Act. | Dyn. | Env. |
| SAC | $8168 \pm 4702$ | $10108 \pm 5556$ | $11099 \pm 4909$ |
| Robust SAC | $11620 \pm 6410$ | $12597 \pm 5633$ | $14694 \pm 2728$ |
| SAC+OSI | $5754 \pm 1771$ | $7790 \pm 1112$ | $9265 \pm 1064$ |
| SAC+Online LMS | $8703 \pm 5832$ | $9687 \pm 5554$ | $12839 \pm 3369$ |
| **SAC + Ours** | $\mathbf{13981 \pm 5624}$ | $\mathbf{15282 \pm 5979}$ | $\mathbf{19825 \pm 7932}$ |

of post-training disturbances without requiring fault-specific tuning or training-time exposure.

**Performance Under Post-Training Faults.** Table 1 reports mean episodic return at a moderate severity for each fault family on HalfCheetah-v5 and Ant-v5. While several specialized baselines perform well in isolated settings (e.g., SAC+OSI on HalfCheetah-v5 or CMAC under Ant-v5 environment faults), their performance does not transfer consistently across environments or fault categories. In contrast, the proposed inference-time residual controller achieves the strongest overall performance, ranking first or second in all evaluated settings across both environments. It consistently outperforms alternative inference-time residual learning methods and remains competitive with the privileged SAC+OSI baseline, despite operating without system identification, latent fault conditioning, or prior exposure to deployment-time fault distributions.

**Whole-Body Locomotion with Humanoid-v5.** We evaluate inference-time fault recovery on Humanoid-v5, a high-dimensional whole-body locomotion task where post-training faults directly disrupt balance and coordination. Table 2 reports performance under representative actuator, dynamic, and environmental faults. The proposed inference-time residual controller achieves the highest performance across all fault families, outperforming both training-time robustness and alternative online adaptation baselines without requiring retraining or system identification. Full sever-

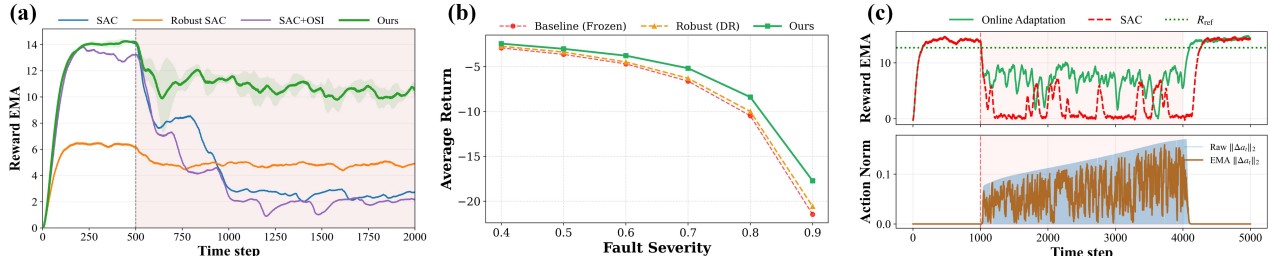

*Figure 2.* **(a)** Inference-time residual adaptation under unanticipated post-deployment faults. A delayed out-of-distribution fault (shaded region) defeats robust training: Robust SAC and SAC+OSI fail, while the proposed method recovers without retraining or exploration. **(b)** Performance on `PandaReach-v3` across fault severities, comparing a frozen baseline, robust training, and the proposed method; inference-time adaptation consistently outperforms both baselines. **(c)** Soft-gated cerebellar residual under actuator bias. *Top*: EMA of episodic reward, showing residual suppression under nominal conditions, activation under sustained degradation, and decay after fault removal. *Bottom*: $\ell_2$ norm of the residual action across joints.

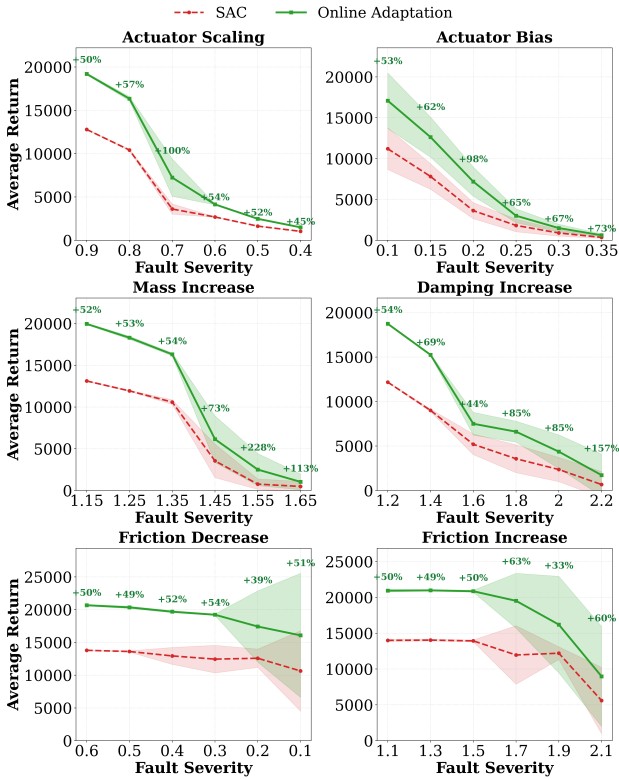

*Figure 3.* Robustness across fault severities on *HalfCheetah-v5*. Episodic return under actuator, dynamic, and environmental perturbations; error bars denote standard deviation over 30 rollouts.

ity sweeps on `Humanoid-v5` and `Ant-v5`, characterizing performance across fault magnitudes and the operating limits of inference-time adaptation, are provided in Appendix B.

**Nominal Performance Preservation.** A key requirement of post-training adaptation is preserving nominal behavior when no actionable fault is present. We evaluate this property using a diagnostic soft-gating mechanism that suppresses residual corrections under nominal conditions. Fig-

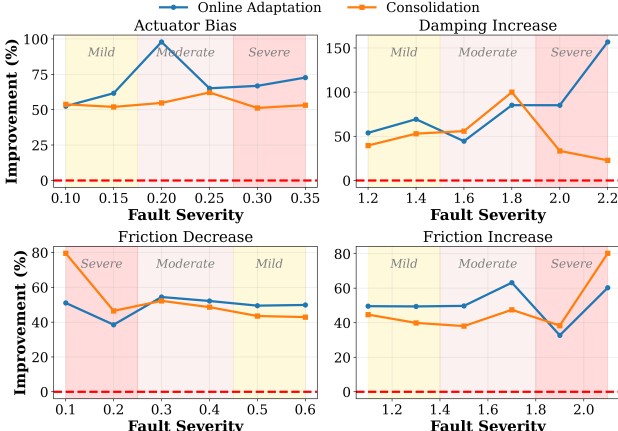

*Figure 4.* Relative improvement over a frozen SAC baseline across fault severities on `HalfCheetah-v5`, comparing inference-time adaptation and policy consolidation under actuator, damping, and friction faults.

ure 2 (c) shows that residual activity remains near zero prior to fault onset while performance tracks the nominal baseline. Following fault injection, sustained degradation triggers corrective activity, resulting in partial recovery. After fault removal, residual activity rapidly decays and performance returns to the nominal regime. Together, these results demonstrate a *do-no-harm* property: the controller intervenes only when sustained performance degradation is detected and remains inactive during nominal operation.

**Unanticipatable Post-Deployment Faults.** Training-time robustness is effective when fault structure is known *a priori*, but becomes unsuitable when faults arise unexpectedly after extended nominal deployment or cannot be safely induced during training. In such regimes, robustness through training-time exposure is fundamentally misaligned with the deployment setting. Figure 2 (a) evaluates this regime using delayed fault injection. Following nominal deployment, a structural disturbance outside the training distribu-

*Table 3.* Ablation results on `HalfCheetah-v5` under representative actuator, dynamic, and environmental faults (mean $\pm$ std over 30 runs). **Bold** indicates best; underlined indicates second-best.

|  | No Gran. Exp. | No Temp. Filt. | No Microzones | No Fast/Slow | No Meta Adapt. | **Ours** |
|---|---|---|---|---|---|---|
| Actuator | $6935 \pm 1968$ | $6216 \pm 1623$ | $6897 \pm 1668$ | $7100 \pm 2263$ | $5843 \pm 1031$ | **$7239 \pm 2159$** |
| Dynamic | $7140 \pm 1515$ | $7468 \pm 1494$ | $7182 \pm 1218$ | $7132 \pm 1313$ | $7116 \pm 1603$ | **$7502 \pm 1281$** |
| Environment | $19415 \pm 3723$ | $19314 \pm 4258$ | $19031 \pm 4533$ | **$19567 \pm 3731$** | $18973 \pm 4054$ | $19517 \pm 3834$ |

tion is introduced, leading to a sharp and sustained performance collapse for training-time robustness methods. In contrast, the proposed inference-time residual controller adapts online and recovers a substantial fraction of nominal performance without retraining or prior fault exposure, motivating inference-time adaptation for *unanticipatable post-deployment faults*.

**Extension Beyond Locomotion.** To evaluate generality beyond cyclic locomotion, we apply the cerebellar residual controller to a non-cyclic manipulation task under actuator faults (Fig. 2 (b)). Phase-aligned reference trajectories are omitted, microzones are indexed by normalized task progress, and the error signal is defined by goal-directed end-effector deviation rather than joint-space tracking error. All other architectural components, including fixed granule feature expansion, dual-timescale residual learning, and confidence-based meta-adaptation, remain unchanged, demonstrating inference-time fault recovery in manipulation without reliance on periodic structure or precomputed reference motion. Additional implementation and evaluation details are provided in Appendix G.

**Ablation Studies.** Table 3 reports ablation results on `HalfCheetah-v5` under representative actuator, dynamic, and environmental faults, evaluating the contribution of key architectural components, including granule feature expansion, temporal filtering, phase-structured microzones, multi-timescale residual learning, and confidence-based meta-adaptation. Removing any individual component reduces performance relative to the full method, indicating that robust inference-time fault recovery arises from the combined effect of multiple complementary mechanisms.

Figure 5 evaluates sensitivity to reference-trajectory design. Phase-indexed reference lookup outperforms time-indexed lookup under increasing damping and friction faults, with fixed phase offsets ($\pm 10\%$, $\pm 15\%$) inducing moderate, non-monotonic performance variation across severities. Removing the feedforward reference acceleration ($\ddot{q}_d$) degrades performance under severe dynamic perturbations. Full ablation details, including extended reference sensitivity and architectural design analyses, are provided in the Appendix D.1.

**Inference-Time Adaptation and Policy Consolidation.** We compare inference-time residual adaptation and policy consolidation on `HalfCheetah-v5` across representative actuator, dynamic, and environmental faults (Fig. 4).

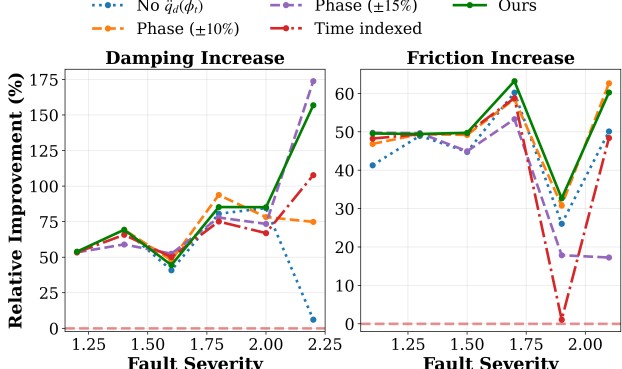

*Figure 5.* Reference-trajectory ablations on *HalfCheetah-v5* under damping and friction increase. Shown are relative performance improvements for the full method, removal of phase-indexed reference acceleration, fixed phase offsets ($\pm 10\%$, $\pm 15\%$), and time-indexed reference lookup, evaluated across fault severities.

Inference-time adaptation consistently improves performance across all tested severities, enabling reliable recovery without modifying the frozen base policy.

Under sustained faults, the residual controller learns structured, persistent corrections that can be consolidated into a separate static policy adapter. Consolidation is performed independently for each fault severity, reflecting the severity-specific structure of the learned residuals, and allows the cerebellar pathway to release plasticity once compensation is absorbed. This yields robust performance across all fault families without requiring continued online adaptation. These results highlight complementary regimes: inference-time adaptation provides immediate, conservative recovery, while consolidation transfers persistent fault structure into a static adapter for sustained robustness. Implementation details and the consolidation protocol are described in Appendix F.

# 7. Conclusions

We address post-training fault recovery in robotic control settings where retraining, exploration, or explicit system identification are impractical. We show that a cerebellar-inspired residual controller with performance-driven meta-adaptation enables effective inference-time recovery while preserving the nominal behavior of a frozen reinforcement learning policy. The residual pathway operates conservatively, remaining inactive under nominal conditions

and providing bounded, error-driven correction only when sustained degradation is detected, yielding a clear *do-no-harm* property across `HalfCheetah-v5`, `Ant-v5`, and `Humanoid-v5`. For persistent faults, structured residual corrections can be consolidated into lightweight static policy adapters, transferring fault structure into a linear parameterization and freeing residual plasticity. By unifying conservative inference-time adaptation with offline absorption of persistent structure, this framework bridges adaptive control and reinforcement learning and enables deployment-time robustness in high-dimensional locomotion tasks.

## Impact Statement

This work studies methods for improving the robustness of reinforcement learning based control systems under non-stationary dynamics and post-training faults. Such advances may contribute to more reliable learning-based controllers in simulated and real-world robotic settings.

All experiments are conducted in simulation, and the proposed methods are not intended for direct deployment in safety-critical systems without additional validation. While improved adaptability can benefit applications in robotics and autonomous systems, inappropriate or premature deployment of adaptive control mechanisms may introduce safety risks if system behavior is not carefully monitored.

We do not anticipate immediate negative societal impacts arising from this work. The primary contribution is methodological, advancing understanding of inference-time adaptation and residual control in reinforcement learning.

## Acknowledgements

This work was supported by the National Science Foundation (NSF) award CCF-2106964 and NSF EFRI BRAID Program under award EFMA-2317974.

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

# A. Implementation Details

This appendix provides implementation-level details required for reproducibility. All descriptions apply to `HalfCheetah-v5` unless stated otherwise; experiments on `Ant-v5` and `Humanoid-v5` reuse the same architecture with minor parameter scaling. Algorithm 1 summarizes the complete inference-time procedure, showing how phase-indexed reference lookup, local residual learning, and performance-driven meta-adaptation interact within a single closed-loop system. The procedure can be viewed as a two-level closed-loop adaptive system: an inner loop performs local error-driven residual learning, while an outer meta-controller regulates residual gain and plasticity using sustained reward trends.

## A.1. Phase-Indexed Reference Trajectory

A nominal reference trajectory is generated once offline by rolling out the frozen base policy in the fault-free environment. Joint positions $q_d$, velocities $\dot{q}_d$, and phase estimates $\phi \in [0, 1)$ are recorded at each timestep. Desired accelerations $\ddot{q}_d$ are computed offline via finite differencing of $\dot{q}_d$ using the environment timestep.

At inference time, the reference trajectory is indexed by phase rather than timestep. The current phase $\phi_t \in [0, 1)$ is estimated online from joint kinematics using a deterministic phase-portrait estimator. For locomotion tasks, we compute phase from a single dominant joint coordinate. Given the selected joint position $q_i$ and velocity $\dot{q}_i$,

---

*Algorithm 1.* Inference-Time Cerebellar Residual Control

**Require:** Frozen policy $\pi_\theta$, reference $\mathcal{D}_{\text{ref}}$, fixed feature matrix $V$, microzones $k = 1, \ldots, K$
1: Initialize $W_k^{\text{fast}}, W_k^{\text{slow}} \leftarrow 0$ for all $k$
2: Initialize traces $\Phi^E, \Phi^I \leftarrow 0$, gain $g_0$
3: Initialize $\bar{R} \leftarrow 0$, $\bar{R}^\star \leftarrow -\infty$
4: **for** each deployment step $t$ **do**
5:      Observe $s_t$ and $(q_t, \dot{q}_t)$
6:      $a_t^{\text{nom}} \leftarrow \pi_\theta(s_t)$
7:      Estimate phase $\phi_t$ from joint kinematics
8:      Retrieve $(q_d, \dot{q}_d, \ddot{q}_d)(\phi_t)$ from $\mathcal{D}_{\text{ref}}$
9:      $r_t \leftarrow (\dot{q}_d - \dot{q}_t) + \Lambda(q_d - q_t)$
10:     $h_t \leftarrow \sigma(V[q_t, \dot{q}_t, \ddot{q}_d])$
11:     Update $\Phi^E, \Phi^I$ and set $\Phi_t \leftarrow \Phi^E - \Phi^I$
12:     Compute microzone weights $w_k(\phi_t)$
13:     $a_t^{\text{res}} \leftarrow g_t \sum_k w_k(\phi_t)(W_k^{\text{fast}} + W_k^{\text{slow}})\Phi_t$
14:     Clip $a_t^{\text{res}}$ and apply directional gate
15:     Execute $a_t \leftarrow a_t^{\text{nom}} + a_t^{\text{res}}$
16:     Observe reward $R_t$
17:     **for** active microzones $k$ **do**
18:          **if** $\|r_t\|$ exceeds deadzone **then**
19:               Update $W_k^{\text{fast}}, W_k^{\text{slow}}$ using local plasticity
20:               Project weights to bounded set
21:          **end if**
22:     **end for**
23:     Update $\bar{R}$ and $\bar{R}^\star$
24:     **if** degradation or stagnation is detected **then**
25:          Increase residual gain/plasticity within bounds
26:     **else**
27:          Decay residual gain/plasticity toward baselines
28:     **end if**
29: **end for**

30: **Optional:** for consolidation, fit a static adapter offline from stable slow-pathway residuals.

---

$$\phi_t = \frac{\text{atan2}(\beta \dot{q}_i, \, q_i - \bar{q}_i) + \pi}{2\pi},$$

where $\bar{q}_i$ is an exponential moving average of the joint coordinate and $\beta$ scales velocity to match the position range. The resulting phase is smoothed using a circular exponential moving average to preserve temporal continuity.

Reference states $(q_d, \dot{q}_d, \ddot{q}_d)$ are retrieved by nearest-neighbor lookup over the precomputed reference phases, using a circular distance metric. No interpolation, warping, or online reference regeneration is performed.

The selected phase joint is fixed per environment: `bthigh` (q[0], qd[0]) for `HalfCheetah-v5`, `hip_z` (q[4], qd[4]) for `Ant-v5`, and `abdomen_z` (q[5], qd[28]) for `Humanoid-v5`. These joints were selected because they exhibit stable periodic structure under the nominal policy. A single dominant joint was sufficient in our experiments; using additional joints did not improve performance.

## A.2. Random Features and Temporal Filtering

The cerebellar module uses a fixed random feature expansion of dimension $M = 2500$. Feature weights are sampled once from a zero-mean Gaussian with standard deviation $0.04$ and remain fixed throughout deployment. Inputs $x_t = [q_t, \dot{q}_t, \ddot{q}_d(\phi_t)]$ are projected via $h_t = \text{ReLU}(V x_t)$.

Temporal structure is captured using dual exponential filters,

$$\Phi_E \leftarrow \Phi_E + \alpha_E(h_t - \Phi_E), \quad \Phi_I \leftarrow \Phi_I + \alpha_I(h_t - \Phi_I),$$

with $\alpha_E = \Delta t/0.03$ and $\alpha_I = \Delta t/0.30$. The effective feature activity is $\Phi_t = \Phi_E - \Phi_I$. This provides short- and long-timescale eligibility traces without recurrence or backpropagation through time.

### A.3. Microzones and Fast/Slow Residual Heads

The residual controller is partitioned into $K$ phase-conditioned microzones, where each microzone is a local residual module that restricts adaptation to a region of the trajectory. Each microzone maintains a fast head $W_k^{\text{fast}}$ and a slow head $W_k^{\text{slow}}$. At timestep $t$, the phase $\phi_t$ activates a small subset of microzones through smooth weights $w_k(\phi_t)$, and the residual is computed as a weighted blend of the corresponding local residual outputs.

The fast head adapts rapidly to transient disturbances and includes mild decay, while the slow head accumulates persistent corrections without decay. A minimum microzone weight is enforced to prevent inactive zones. This separation enables transient correction, long-term compensation, and later consolidation while preserving a linear-in-parameters form.

### A.4. Local Learning and Stability Measures

Both fast and slow heads are updated online using a normalized LMS-style local rule driven by the composite tracking error $r_t$. Updates are suppressed when $\|r_t\|$ falls below a fixed deadzone threshold to prevent parameter drift under near-nominal conditions.

Residual outputs are clipped elementwise, and a directional-consistency gate suppresses residual actions whose global dot product with the base policy action is negative. The gate is applied globally (not per joint) on unnormalized action vectors, preserving coordinated cross-joint structure. While we do not provide formal stability guarantees, several design choices empirically promote bounded behavior: normalized updates, residual clipping, deadzones, and conservative meta-gating of residual gain. Across all experiments, cerebellar weights remain bounded and no instability or divergence was observed.

### A.5. Meta-Adaptation

A meta-controller monitors an exponential moving average (EMA) of the task reward with smoothing factor $\rho = 0.1$ over a 50-step window. Two events trigger adaptation: (i) performance degradation, defined as an EMA drop below $-0.3$ relative to the historical maximum, and (ii) stagnation, defined as no improvement for 100 consecutive steps.

Upon event detection, residual confidence, learning rates, gain, and tracking-error weights are smoothly relaxed toward conservative targets within predefined bounds. In the absence of events, all parameters decay toward nominal baselines, preventing abrupt changes and preserving nominal behavior.

We observed moderate sensitivity to the reward-drop threshold and stagnation horizon, but the qualitative recovery trends were preserved across the tested settings. Overly tight reward-drop thresholds can cause premature gain escalation and oscillatory corrections, while very short stagnation horizons can trigger adaptation before the slow residual head stabilizes. These effects are mitigated by residual clipping, deadzones, directional gating, and bounded parameter updates.

### A.6. Hyperparameters and Scaling

Table 5 lists all hyperparameters. Parameters were initialized on `HalfCheetah-v5` and held fixed across all fault types and severity levels. For `Ant-v5` and `Humanoid-v5`, only base learning-rate, gain, and confidence scales were adjusted to account for differences in actuation magnitude and reward scale. No per-fault or per-severity tuning was performed.

### A.7. Computational Overhead

We profile the inference-time cost of the residual controller on a single CPU thread. Runtime is measured per control step after a 200-step warmup and averaged over 2000 rollout steps. Environment stepping is excluded, so the reported values measure only policy and controller computation. The residual-controller timing includes phase estimation, feature projection, excitatory/inhibitory temporal filtering, soft microzone readout, residual action computation, and the local normalized weight update.

The residual controller adds a small constant computational overhead. Although the relative overhead is large compared to the very small SAC-only forward pass, the absolute cost remains below 2 ms per control step in all evaluated environments. These results suggest that the proposed inference-time adaptation is unlikely to create a practical computational bottleneck in our evaluated settings.

| Environment | SAC only (ms) | SAC + residual (ms) | Overhead |
|---|---|---|---|
| HalfCheetah-v5 | 0.12 | 0.46 | +0.35 ms |
| Ant-v5 | 0.14 | 0.43 | +0.28 ms |
| Humanoid-v5 | 0.15 | 1.70 | +1.55 ms |

*Table 4.* Inference-time profiling per control step, averaged over 2000 rollout steps after a 200-step warmup, on a single CPU thread (Intel i7-13700, PyTorch 2.3, SB3 2.4.1). Runtime includes the SAC policy forward pass and the residual-controller computation. Environment stepping is excluded.

# B. Additional Experimental Results on Ant and Humanoid

This appendix reports additional severity-sweep evaluations on higher-dimensional locomotion systems. These environments exhibit stronger dynamical coupling and coordination requirements, providing a complementary assessment of inference-time residual adaptation beyond `HalfCheetah-v5`.

## B.1. Ant-v5 Severity Sweeps

We report severity sweeps on `Ant-v5` under actuator scaling, joint damping increase, and friction increase faults. These fault families capture actuator-level, dynamical, and environmental perturbations commonly encountered after policy training.

### B.1.1. RESULTS AND INTERPRETATION

Figure 6 summarizes performance across fault severities. Inference-time residual adaptation improves episodic return relative to the frozen SAC baseline across all evaluated actuator scaling severities. For joint damping increase and friction increase faults, performance gains are observed across mild to moderate severities, with degradation occurring only at the most severe levels.

Overall, these results indicate that inference-time residual adaptation remains effective on `Ant-v5` across a broad range of fault magnitudes, while highlighting expected limits under extreme whole-body perturbations that challenge coordinated multi-legged locomotion.

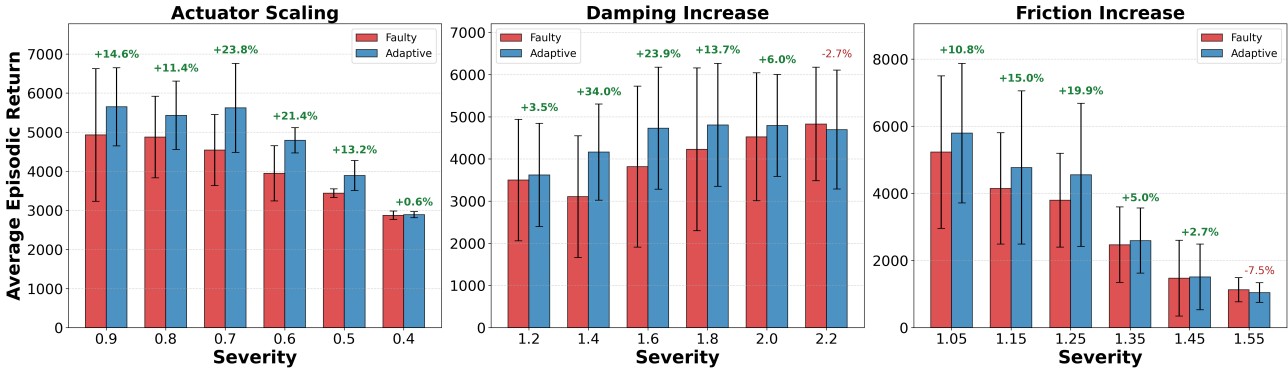

*Figure 6.* Severity sweeps on `Ant-v5` under actuator scaling, joint damping increase, and friction increase faults. Inference-time residual adaptation improves performance across mild to moderate severities relative to a frozen SAC baseline, with degradation observed only at some of the most severe fault levels. Error bars denote one standard deviation.

## B.2. Humanoid-v5 Severity Sweeps

We conduct severity sweep evaluations on `Humanoid-v5` under actuator scaling, joint damping increase, and friction increase faults. These experiments complement the main results by characterizing how inference-time residual adaptation behaves across a range of fault magnitudes in high-dimensional whole-body locomotion.

| Category | Parameter | Value |
|---|---|---|
| Cerebellar Residual | Input dimension | 18 |
| | Random feature dimension $M$ | 2500 |
| | Excitatory filter rate $\alpha_E$ | $\Delta t/0.03$ |
| | Inhibitory filter rate $\alpha_I$ | $\Delta t/0.30$ |
| | Residual clip $\|\tau\|_\infty$ | 0.15 |
| | Deadzone threshold | 0.25 |
| | Momentum coefficient | 0.85 |
| | $\ell_2$ regularization $\gamma$ | $4 \times 10^{-6}$ |
| | Per-joint learning rate $\eta$ | per-joint (fixed) |
| | Tracking error gain $\lambda$ | per-joint (fixed) |
| | Residual scaling gain $g$ | 0.35 |
| | Learning start step | 200 |
| Phase & Microzones | Phase variable range | $[0, 1)$ |
| | Phase smoothing factor | 0.90 |
| | Number of microzones $K$ | 4 |
| | Microzone weighting | Soft (Gaussian) |
| | Minimum zone weight | 0.05 |
| | Phase reference indexing | Nearest neighbor |
| | Phase estimator | Phase portrait ((atan2-based)) |
| Fast / Slow Heads | Fast learning-rate scale | $5.0\times$ base |
| | Slow learning-rate scale | $0.2\times$ base |
| | Fast decay rate | $10^{-3}$ |
| | Slow decay rate | 0 (no decay) |
| | Fast head role | Transient correction |
| | Slow head role | Persistent bias |
| Meta-Adaptation | Initial confidence $c_0$ | 0.40 |
| | Confidence bounds | $[0, 0.70]$ |
| | Reward EMA factor $\rho$ | 0.10 |
| | Reward window size | 50 |
| | Performance check frequency | 20 steps |
| | Stagnation threshold | 100 steps |
| | Reward drop threshold $\delta_{\text{drop}}$ | $-0.30$ |
| | Learning-rate multiplier range | $[0.5, 2.0]$ |
| | Gain multiplier range | $[0.5, 2.0]$ |
| | Lambda multiplier range | $[0.7, 1.5]$ |

*Table 5.* Design parameters for the phase-structured cerebellar residual controller and meta-adaptation mechanism. Unless otherwise stated, parameters were initialized on `HalfCheetah-v5` and lightly scaled for `Ant-v5` and `Humanoid-v5` to account for differences in actuation scale and reward dynamics. Per-joint base learning rates and tracking gains are fixed and shared across all fault types and severities.

### B.2.1. FAULT APPLICATION PROTOCOL AND SCOPE

For `Humanoid-v5`, actuator scaling and joint damping faults are applied to a fixed subset of lower-body actuated degrees of freedom (hips, knees, and ankles), which dominate propulsion and balance during locomotion. Applying comparable faults uniformly across all actuators leads to near-immediate collapse for both frozen and adaptive policies, yielding near-zero rewards and preventing meaningful evaluation of recovery dynamics. The residual controller itself still operates over the full action space; the restricted fault set is used only to define a recoverable but challenging evaluation regime.

The impacted subset is selected based on anatomical role rather than task-specific tuning, and is held fixed across all baselines and fault types to ensure fair comparison. While different lower-body subsets may induce varying degrees of difficulty, our goal is not to exhaustively enumerate all possible fault patterns, but to characterize the operating regime in which inference-time residual adaptation is effective for localized post-training degradation. Exploring full combinatorial coverage of actuator subsets remains an interesting direction for future work.

Environmental friction increase is applied globally by scaling the MuJoCo contact friction parameters, affecting all ground interactions uniformly. All fault configurations are applied identically for frozen, training-time robust, and inference-time adaptive methods.

### B.2.2. RESULTS AND INTERPRETATION

Figure 7 reports severity sweeps for each fault family. Inference-time residual adaptation consistently improves performance across mild to moderate fault severities for all evaluated fault types, demonstrating that local, error-driven correction can effectively compensate for a range of post-training disturbances in whole-body locomotion.

At higher severities, performance degrades relative to the frozen baseline for a subset of fault configurations. These regimes typically correspond to perturbations that induce global coordination or balance failures, exceeding the corrective capacity of purely local joint-level residual adaptation. Together, these results delineate the operating regimes in which inference-time residual learning is effective for humanoid locomotion, and characterize the limits of local adaptation under severe whole-body disruptions.

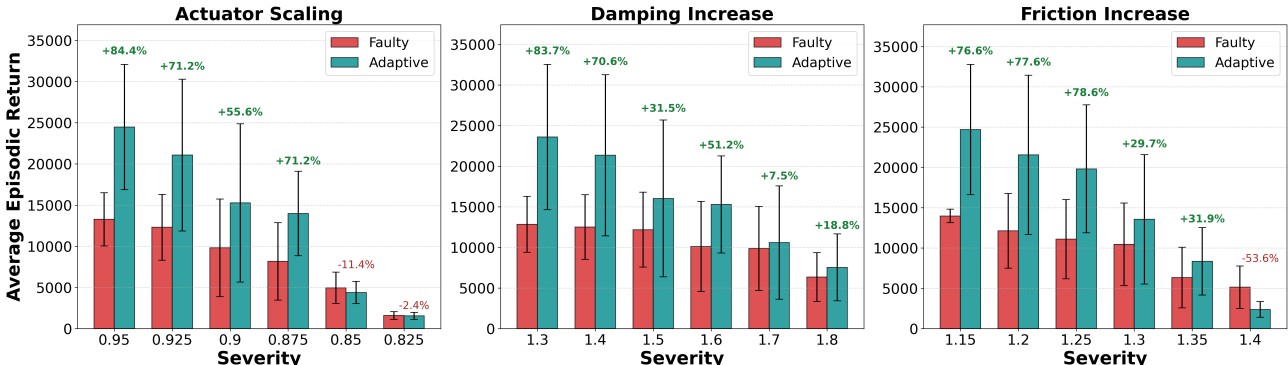

*Figure 7.* Severity sweeps on `Humanoid-v5` under actuator scaling, joint damping increase, and friction increase faults. Inference-time residual adaptation consistently improves performance across mild to moderate fault severities, while degradation occurs only at some severe perturbation levels that induce global coordination and balance failures. Error bars denote one standard deviation.

## C. Additional Baselines Details

### C.1. Robust SAC Baseline

Robust SAC is implemented using domain randomization during training. A custom environment wrapper applies randomized actuator, dynamics, and environmental faults at the beginning of each episode. Faults are sampled independently per episode and persist for the full rollout.

At reset, the environment is restored to nominal physics before sampling a fault. With probability 0.2, the episode is nominal; with probability 0.8, a fault type is sampled uniformly from actuator scaling, actuator delay, body mass increase, joint damping increase, and ground friction variation. Fault severities are sampled uniformly over the same ranges used during evaluation. Actuator delays are implemented using a FIFO action buffer, while physics faults modify MuJoCo model parameters directly.

The Robust SAC policy uses the same network architecture, optimizer settings, and entropy tuning as the nominal SAC baseline. No fault labels are provided to the policy, and hyperparameters are not tuned per fault type.

While Robust SAC improves average performance across randomized training conditions, we observe that it can underperform the nominal SAC policy under certain post-training fault scenarios. This behavior is consistent with known robustness–performance tradeoffs in domain-randomized policies, where invariance across a wide range of disturbances may reduce sensitivity to localized or structured perturbations.

The same robust training protocol is applied to `Ant-v5` and `Humanoid-v5`, with increased training budgets to account for higher state and action dimensionality. Robust SAC is trained for $3 \times 10^6$ steps on `Ant-v5` and $5 \times 10^6$ steps on `Humanoid-v5`. For each environment, fault families, severity ranges, and affected joints are defined to match the corresponding evaluation settings, while all optimization hyperparameters are kept fixed across environments.

## C.2. SAC + OSI Baseline

This appendix describes the Soft Actor–Critic with Online System Identification (SAC+OSI) baseline used for comparison. SAC+OSI addresses deployment-time disturbances by explicitly estimating latent system parameters and conditioning the control policy on these estimates. Importantly, SAC+OSI performs no online policy or parameter learning at inference time.

### C.2.1. LATENT DYNAMICS FORMULATION

We consider a family of environments parameterized by an unobserved latent dynamics vector $z \in \mathcal{Z}$. The resulting Markov decision process is

$$\mathcal{M}(z) = (\mathcal{S}, \mathcal{A}, p_z(s_{t+1} \mid s_t, a_t), r(s_t, a_t)),$$

where $p_z$ captures changes in transition dynamics induced by post-training faults, such as actuator degradation or modified physical parameters.

### C.2.2. DYNAMICS-CONDITIONED POLICY

Unlike a standard policy $\pi(a \mid s)$, SAC+OSI trains a policy explicitly conditioned on the latent dynamics: $a_t \sim \pi_\theta(\cdot \mid s_t, z)$ Conditioning is implemented by augmenting the observation with $z$, $\tilde{s}_t = [s_t; z]$, and training a single policy capable of representing behaviors across a family of faulted dynamics. The conditioned policy is trained offline using standard Soft Actor–Critic under domain randomization over $z$.

### C.2.3. FAULT PARAMETERIZATION

Each fault is parameterized by a discrete fault family $f \in \{1, \dots, F\}$ and a continuous severity parameter $\theta$. The latent vector is defined as

$$z = [\text{onehot}(f); \theta_{\text{norm}}] \in \mathbb{R}^{F+1},$$

where $\text{onehot}(f)$ encodes the fault type and

$$\theta_{\text{norm}} = \text{clip}\left( \frac{\theta - \theta_{\min}^{(f)}}{\theta_{\max}^{(f)} - \theta_{\min}^{(f)}}, 0, 1 \right)$$

is a normalized severity value based on the domain randomization range used during training.

### C.2.4. ONLINE SYSTEM IDENTIFICATION (OSI)

At deployment, the latent parameter $z$ is unknown and must be inferred from interaction data. SAC+OSI employs a separate Online System Identification network $g_\phi$ that predicts $z$ from a fixed-length history window

$$w_t = \big[ (s_{t-K}, a_{t-K}), \dots, (s_{t-1}, a_{t-1}) \big].$$

The OSI prediction is

$$\hat{z}_t = g_\phi(w_t).$$

The OSI network is trained offline using supervised learning, leveraging the fact that $z$ is known during data collection. The training objective combines a cross-entropy loss over fault families and a mean squared error loss over normalized severity:

$$\mathcal{L}_{\text{OSI}} = \mathcal{L}_{\text{type}} + \lambda_{\text{sev}} \, \mathcal{L}_{\text{sev}},$$

with $\lambda_{\text{sev}} = 2$ in all experiments to balance the scale of the regression and classification losses, and is not tuned per environment.

### C.2.5. TRAINING COVERAGE AND BUDGET

Both the conditioned policy and the OSI network are trained with explicit access to fault identities and severity ranges during data collection. For each environment, fault families and severity ranges match those used for evaluation in that environment.

The conditioned SAC policy is trained for $3 \times 10^6$ environment steps, while the OSI network is trained for 25 epochs on a fixed offline dataset of state–action histories collected under randomized fault conditions. The training budget of SAC+OSI exceeds that of the nominal SAC and robust SAC baselines to account for the additional complexity introduced by latent conditioning.

INFERENCE-TIME EXECUTION

At inference time, both the conditioned policy $\pi_\theta$ and the OSI estimator $g_\phi$ are frozen. At each time step,

$$\hat{z}_t = g_\phi(w_t), \qquad a_t \sim \pi_\theta(\cdot \mid s_t, \hat{z}_t).$$

No policy gradients, replay buffers, or online parameter updates are performed. Adaptation occurs solely through changes in the estimated latent parameter $\hat{z}_t$.

**C.3. SAC + Offline Residual Network Baseline**

This section provides implementation details for the offline residual network baseline used in both HalfCheetah-v5 and Ant-v5 experiments.

C.3.1. BASELINE DEFINITION

The offline residual baseline augments a frozen SAC policy with an additive corrective action produced by a neural network trained entirely offline. The executed action at time $t$ is given by

$$a_t = a_t^{\text{SAC}} + c \cdot g \cdot \tau_t^{\text{NN}},$$

where $a_t^{\text{SAC}}$ denotes the frozen policy output, $\tau_t^{\text{NN}}$ is the residual predicted by the offline network, and $c$ and $g$ are fixed confidence and gain parameters. These parameters are held constant across all environments, fault types, and severities.

To ensure consistency across baselines, residual outputs are clipped to a fixed magnitude, matching the residual injection mechanism used by the proposed method.

C.3.2. INPUT REPRESENTATION

At each timestep, the offline residual network receives instantaneous joint-level information and reference signals:

$$x_t = \left[q_t,\ \dot{q}_t,\ \ddot{q}_{ref,t}\ e_t,\ \dot{e}_t\right],$$

where $q_t$ and $\dot{q}_t$ denote joint positions and velocities, $\ddot{q}_{ref,t}$ is a reference joint acceleration derived from a nominal rollout of the base policy, and $e_t = q_{ref,t} - q_t$, $\dot{e}_t = \dot{q}_{ref,t} - \dot{q}_t$ are tracking errors.

This representation provides access to instantaneous feedforward and feedback signals, while remaining memoryless and fully feedforward.

C.3.3. SUPERVISION AND TRAINING TARGETS

The offline residual network is trained using a fixed proportional–derivative (PD) control law as supervision. Training targets are defined as

$$\tau_t^{\text{target}} = K_p e_t + K_d \dot{e}_t,$$

where $K_p$ and $K_d$ are fixed per-joint gains chosen conservatively to avoid destabilizing the base policy. Target residuals are clipped to the same bounds used during evaluation.

This design yields a static mapping from instantaneous tracking errors to corrective torques, without incorporating inference-time adaptation, temporal credit assignment, or performance-driven modulation.

C.3.4. OFFLINE TRAINING PROCEDURE

Training data are collected by rolling out the frozen SAC policy under post-training faults, without applying any residual correction during data collection. To promote generalization, fault severities used during training span a wider range than those used at evaluation.

The residual network is trained offline using a Huber (Smooth L1) loss to regress the PD targets. Once training is complete, all network parameters are frozen. No retraining, fine-tuning, or parameter updates occur during deployment.

### C.3.5. DEPLOYMENT CHARACTERISTICS

At inference time, the offline residual network operates as a fixed, feedforward compensator. In particular:

- no online learning or weight updates are performed,
- no reward monitoring or performance-based gating is used,
- confidence and gain parameters remain constant throughout the episode.

As a result, the offline residual baseline applies identical correction strategies regardless of fault severity, duration, or temporal evolution.

### C.3.6. SCOPE AND LIMITATIONS

The offline residual baseline provides a strong reference for evaluating static, pre-trained compensation mechanisms. While such models can partially mitigate performance degradation under moderate disturbances, they lack the ability to regulate residual gain or adapt to changing fault conditions. In particular, they do not distinguish between transient and persistent faults, nor do they suppress intervention when nominal behavior is preserved.

## C.4. SAC + CMAC Residual Control Baseline

We include a SAC + CMAC baseline to compare against a classical function approximation based residual controller with online adaptation. This baseline augments a frozen Soft Actor-Critic (SAC) policy with a Cerebellar Model Articulation Controller (CMAC) style residual module, representing a commonly used approach in adaptive control and robotics.

### C.4.1. ARCHITECTURE.

The base SAC policy is trained under nominal, fault-free dynamics and remains frozen during evaluation. A linear residual controller operates in parallel, producing an additive correction to the SAC action. The residual is parameterized by a fixed set of random ReLU features (analogous to coarse CMAC tilings or random-feature approximations), followed by a linear readout. The feature projection is initialized once and remains fixed throughout deployment.

### C.4.2. LEARNING RULE.

Residual weights are updated online using a normalized least-mean-squares (NLMS) rule driven by joint-level tracking error,

$$r_t = \dot{e}_t + \lambda e_t,$$

where $e_t$ denotes deviation from a reference trajectory generated by the frozen policy under nominal dynamics. The learning rate, residual gain, and $\lambda$ vector are fixed hyperparameters and are not adapted during inference.

### C.4.3. DESIGN SCOPE.

The SAC + CMAC baseline uses a standard CMAC-style residual controller with fixed features and online LMS updates. It does not include performance gating, temporal filtering, or meta-adaptation, which are introduced only in the proposed method.

### C.4.4. PURPOSE IN EVALUATION.

The SAC + CMAC baseline isolates the effect of structured, performance-driven inference-time adaptation. By comparing against a residual controller with fixed random features and static LMS updates, we assess whether the gains of the proposed method arise from cerebellar-inspired structure and conservative meta-adaptation rather than residual learning alone.

## C.5. SAC + Online LMS Baseline

This section describes the SAC + Online LMS baseline used to evaluate naive inference-time residual learning without cerebellar structure or meta-adaptation.

### C.5.1. MOTIVATION AND SCOPE.

The SAC + Online LMS baseline is designed to isolate the effect of *simple online adaptive residual learning* at inference time. Unlike the proposed cerebellar residual controller, this baseline does not employ high-dimensional feature expansion, temporal filtering, confidence gating, or performance-driven meta-adaptation. Instead, it implements a classical *normalized least-mean-squares (NLMS)* update to adapt a linear residual controller online. This baseline answers the question: to what extent can straightforward LMS-style adaptation recover performance under post-training faults?

### C.5.2. RESIDUAL POLICY FORMULATION.

Let $\pi_{\text{SAC}}(s_t)$ denote the frozen base policy trained under nominal conditions. The executed action is given by

$$a_t = \pi_{\text{SAC}}(s_t) + g\,\tau_t,$$

where $g$ is a fixed residual gain and $\tau_t$ is a learned residual correction.

The residual is parameterized as a linear function of the input features: $\tau_t = W_t x_t$, where $W_t \in \mathbb{R}^{6\times d}$ is updated online and $x_t \in \mathbb{R}^d$ is the residual input vector.

### C.5.3. INPUT REPRESENTATION.

To ensure a fair comparison, the LMS baseline is provided access to the same nominal reference trajectory used by the proposed method. Specifically, the residual input is defined as $x_t = [q_t,\ \dot{q}_t,\ \ddot{q}_{d,t}]$, where $q_t$ and $\dot{q}_t$ are the measured joint positions and velocities, and $\ddot{q}_{d,t}$ is the reference joint acceleration derived from a nominal rollout of the frozen SAC policy. This yields an 18-dimensional input vector that matches the footprint of the proposed method's raw input for *HalfCheetah-v5*, while omitting any random feature expansion or temporal filtering.

### C.5.4. LEARNING RULE (NORMALIZED LMS).

Residual weights are adapted online using a standard normalized LMS update:

$$W_{t+1} = W_t + \eta\,\frac{r_t x_t^\top}{\|x_t\|^2 + \varepsilon},$$

where $\eta$ is a fixed learning rate, $\varepsilon > 0$ ensures numerical stability, and

$$r_t = \dot{e}_t + \Lambda e_t$$

is the same composite tracking error signal used in the proposed method, with fixed diagonal gain matrix $\Lambda$.

Learning begins after a short warm-up period to avoid transient effects. All parameters remain fixed throughout deployment: no reward monitoring, learning-rate adaptation, gain modulation, confidence variables, or performance-triggered events are used.

### C.5.5. SAFETY CONSTRAINTS AND PARITY WITH THE PROPOSED METHOD.

For parity with the proposed controller and to prevent trivial instability, the LMS baseline employs residual output clipping to a fixed magnitude and an optional directional consistency gate that suppresses residual corrections opposing the base policy action. These mechanisms are included solely for safety and do not introduce performance-driven adaptation.

### C.5.6. HYPERPARAMETERS AND TUNING.

The learning rate $\eta$, residual gain $g$, and tracking gains $\Lambda$ are tuned once on a small validation subset of HalfCheetah-v5 and then held fixed across all environments, fault types, and severity levels. No per-fault or per-environment tuning is performed.

### C.5.7. LIMITATIONS.

While SAC + Online LMS enables genuine inference-time adaptation, it lacks several properties of the proposed cerebellar residual controller. In particular, it does not perform temporal credit assignment, cannot regulate residual gain based on task performance, and does not suppress adaptation under nominal conditions. As a result, LMS-based residual learning may introduce instability or interference under heterogeneous or severe post-training faults.

# D. Additional Ablation Results

## D.1. Reference Trajectory Usage and Sensitivity

This section describes how the nominal reference trajectory is generated and used under phase-based indexing, and evaluates sensitivity to partial reference removal, reference misalignment, and time-based indexing.

### D.1.1. REFERENCE TRAJECTORY GENERATION AND PHASE-BASED INDEXING

A nominal reference trajectory is generated once by rolling out the frozen SAC policy in the fault-free environment prior to deployment. Joint positions $q_d$, velocities $\dot{q}_d$, and accelerations $\ddot{q}_d$ are recorded over a fixed horizon, where accelerations are computed offline via finite differencing of $\dot{q}_d$. Along this trajectory, a phase variable $\phi \in [0, 1)$ is computed using a deterministic phase-portrait estimator based on a dominant joint angle and velocity, yielding a phase-aligned reference $(q_d(\phi), \dot{q}_d(\phi), \ddot{q}_d(\phi))$.

At inference time, the reference is indexed by the current estimated phase $\phi_t$ via nearest-neighbor lookup in phase space. No online trajectory regeneration, time warping, or synchronization is performed. Deviations between the current system state and the phase-aligned reference appear as instantaneous tracking errors and are handled locally by the residual controller. Episodes are truncated to the length of the reference trajectory for consistency across conditions.

### D.1.2. INPUT-LEVEL REFERENCE ABLATION

We evaluate the role of reference information in the cerebellar input by removing the phase-indexed reference acceleration $\ddot{q}_d(\phi_t)$ from the feature vector. In this ablation, the random feature input is reduced from $[q, \dot{q}, \ddot{q}_d(\phi_t)]$ to $[q, \dot{q}]$, while the tracking error used for weight updates remains unchanged.

As shown in Fig. 8, removing reference acceleration produces performance that is broadly comparable to the full method across actuator faults and non-severe regimes, while exhibiting larger degradation under the most severe dynamic and environmental perturbations. This indicates that reference acceleration is not strictly necessary for effective adaptation in mild-to-moderate conditions, but becomes more useful when the system undergoes stronger dynamics shifts where a coarse predictive scaffold can stabilize correction.

### D.1.3. PHASE OFFSET SENSITIVITY

To assess robustness to reference misalignment under phase-based indexing, we introduce fixed phase offsets at the start of each episode. The phase used for reference lookup is perturbed by a constant offset corresponding to $\pm 10\%$ or $\pm 15\%$ of the phase cycle, and the offset remains fixed throughout the episode.

As shown in Fig. 8, moderate phase offsets maintain competitive performance across actuator faults and moderate severities, while larger offsets induce increased variability and more pronounced degradation at the highest severities, particularly for dynamic and environmental faults. These results suggest that approximate phase alignment is sufficient in typical regimes, while severe faults that alter coordination patterns increase sensitivity to reference misalignment.

### D.1.4. TIME-INDEXED REFERENCE ABLATION

We additionally evaluate a time-indexed variant in which both reference lookup and tracking error computation use timestep $t$ rather than the estimated phase $\phi_t$. This removes phase-based invariance while retaining access to the same nominal trajectory.

As shown in Fig. 8, time indexing is competitive in some regimes but can degrade under conditions where faults induce speed variation or temporal drift relative to the nominal rollout, which reduces alignment when indexing is tied directly to elapsed time rather than the system state.

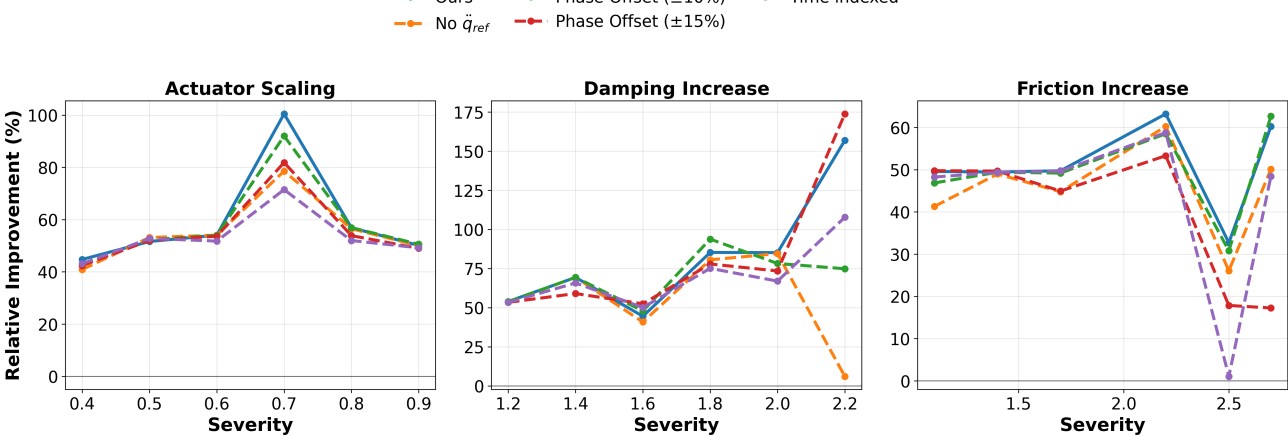

*Figure 8.* Reference usage and indexing ablations. Relative improvement over a frozen SAC baseline for actuator scaling (left), joint damping increase (center), and friction increase (right). Curves correspond to the full method (Ours), removal of reference acceleration input ($\ddot{q}_{\mathrm{ref}}$), time-indexed reference usage, and fixed phase offsets of $\pm10\%$ and $\pm15\%$. Results are reported across fault severities for each fault family.

## D.2. Sensitivity to Meta-Controller Parameters

This section reports an ablation study examining the effect of key meta-controller hyperparameters on inference-time adaptation performance. The purpose of this analysis is to characterize how performance varies under different reward monitoring and adaptation triggering settings, without modifying the underlying residual controller.

### D.2.1. META-CONTROLLER PARAMETERS AND ABLATION SETUP

We consider three components of the meta-controller: (i) the exponential moving average (EMA) smoothing factor used for reward tracking, (ii) the performance drop threshold used to trigger corrective adaptation, and (iii) the stagnation horizon specifying the number of steps without reward improvement before parameter modulation is invoked.

Each parameter is varied independently while all others are held fixed at their default values used in the main experiments. The evaluated ranges are:

- EMA smoothing factor $\alpha \in \{0.05, 0.10, 0.20, 0.30\}$,
- Reward drop threshold $\Delta r \in \{-0.20, -0.30, -0.40, -0.50\}$,
- Stagnation horizon $H \in \{50, 100, 150, 200\}$ steps.

No joint retuning or compensatory adjustments are performed. All experiments are conducted on `HalfCheetah-v5` under actuator scaling faults across multiple severity levels, with results averaged over 30 runs.

### D.2.2. SENSITIVITY TO META-CONTROLLER PARAMETERS

Figure 9 shows relative improvement over the frozen SAC baseline as a function of fault severity for each parameter sweep.

Across the tested settings, the resulting performance curves exhibit similar qualitative trends as severity varies. Differences between parameter choices lead to moderate shifts in performance across conditions, while preserving the overall structure of the severity response. No single parameter setting dominates uniformly across all fault levels.

## D.3. Granule Expansion Size and Directional Gating Ablations

This section analyzes the sensitivity of the proposed method to the granule expansion dimensionality $M$ and the directional gating mechanism. Figure 10 reports relative improvement over a frozen SAC baseline across fault severities for actuator scaling, damping increase, and friction increase.

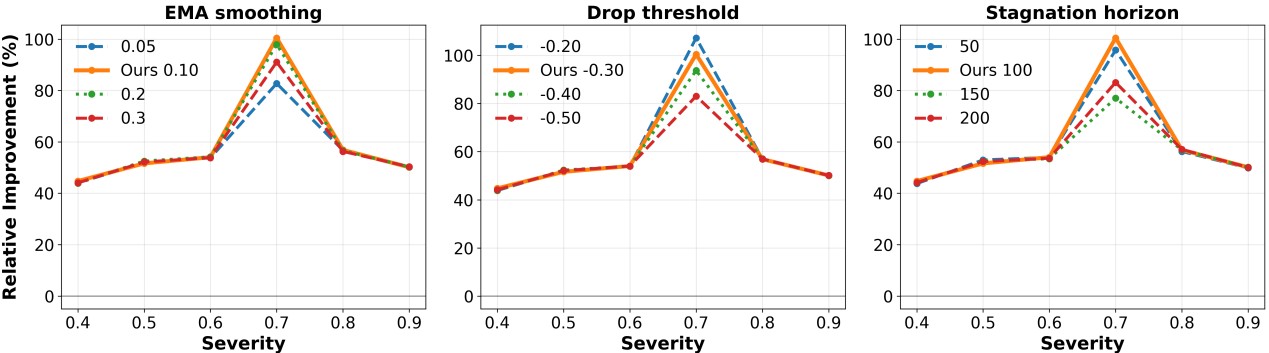

*Figure 9.* Sensitivity of inference-time adaptation to meta-controller parameters under actuator scaling faults on `HalfCheetah-v5`. Left: EMA smoothing factor used for reward tracking. Center: reward drop threshold triggering corrective adaptation. Right: stagnation horizon before exploratory parameter modulation. Relative improvement is reported over a frozen SAC baseline. Results are averaged over 30 runs.

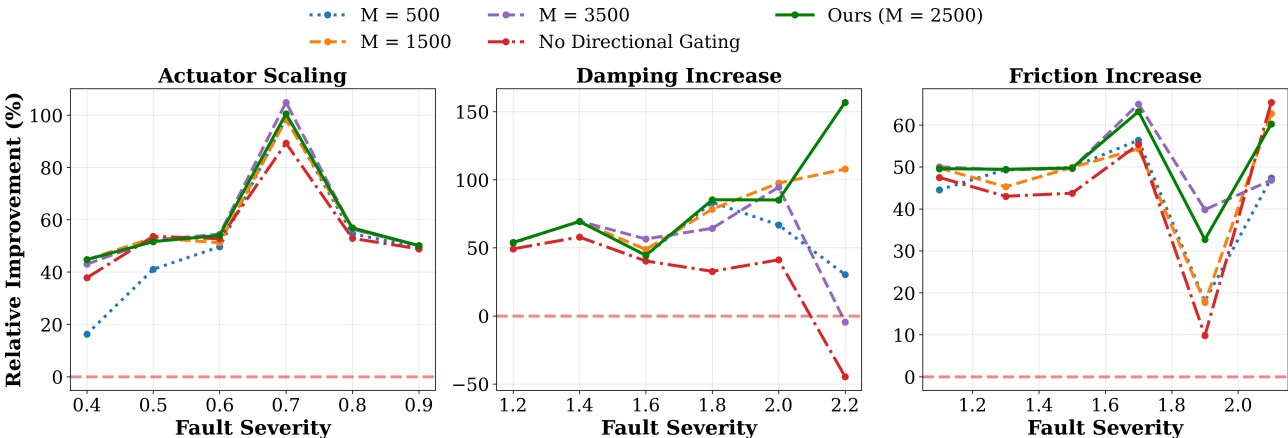

*Figure 10.* Ablation of granule expansion size $M$ and directional gating on `HalfCheetah-v5`. Relative improvement over a frozen SAC baseline is shown across fault severities for actuator scaling, damping increase, and friction increase. Curves compare different granule sizes and the effect of disabling directional gating.

### D.3.1. GRANULE EXPANSION SIZE.

Varying the granule expansion size reveals a trade-off between representational capacity and robustness across fault regimes. Smaller expansions ($M = 500$) can achieve competitive performance at isolated fault severities but fail to do so consistently across fault magnitudes. Moderate expansions ($M \in [1500, 2500]$) provide the most stable improvements across fault families, motivating the choice of $M = 2500$ in the main experiments. Further increasing $M$ yields diminishing returns and, in some regimes, reduced performance, suggesting that larger expansions do not monotonically improve inference-time adaptation.

### D.3.2. DIRECTIONAL GATING.

Disabling directional gating consistently reduces performance across many fault families and severities. While the residual controller remains functional without gating, the resulting corrections are less reliable, leading to weaker recovery. This demonstrates that directional gating plays a stabilizing role by suppressing harmful residual updates while preserving beneficial corrections, supporting conservative inference-time adaptation.

# E. Nominal Performance Preservation

This appendix characterizes the behavior of the proposed residual controller under nominal (no-fault) conditions. While the main paper focuses on robustness to post-training faults, the results here demonstrate that the residual module remains non-interfering when the base policy operates within its nominal performance envelope. No formal safety or stability guarantees are claimed.

## E.1. Residual Policy Formulation

Let $\pi_{\text{SAC}}(s_t) \in \mathbb{R}^6$ denote the frozen base policy trained under nominal conditions. The cerebellar module produces an additive residual correction $\Delta a_t \in \mathbb{R}^6$, yielding the executed action

$$a_t = \pi_{\text{SAC}}(s_t) + g_t\, c_t\, G\, \Delta a_t, \tag{18}$$

where $c_t$ is a bounded confidence variable, $G$ is a fixed residual gain, and $g_t \in [0, 1]$ is a soft gate controlling residual activation.

The residual action is computed using a cerebellum-inspired architecture,

$$\Delta a_t = W_t \Phi_t, \tag{19}$$

where $\Phi_t$ denotes temporally filtered features and $W_t$ is updated online via a local error-driven learning rule.

## E.2. Nominal Reward Baseline

A scalar nominal reward baseline $\bar{R}_{\text{nom}}$ is computed offline by rolling out the frozen SAC policy in the healthy environment across multiple seeds. This baseline is used only for residual gating and analysis and does not affect policy training or execution.

## E.3. Performance-Based Soft Gating

Online performance is monitored using an exponential moving average (EMA) of the reward,

$$\bar{R}_t = \rho R_t + (1 - \rho)\bar{R}_{t-1}. \tag{20}$$

The residual controller is softly gated based on deviations from nominal performance. When $\bar{R}_t$ remains close to $\bar{R}_{\text{nom}}$, the gate $g_t$ stays near zero, suppressing residual intervention. As performance degrades persistently, $g_t$ increases smoothly, allowing the residual pathway to contribute corrective actions.

## E.4. Confidence Dynamics

The confidence variable $c_t$ evolves slowly in response to sustained performance degradation and is bounded for stability. Importantly, confidence alone does not activate the residual controller: effective intervention requires both elevated confidence and a nonzero gate.

## E.5. Do-No-Harm Property

Under nominal conditions,

$$\bar{R}_t \approx \bar{R}_{\text{nom}} \implies g_t \approx 0, \tag{21}$$

which implies

$$a_t \approx \pi_{\text{SAC}}(s_t). \tag{22}$$

Thus, the residual controller remains functionally inactive when the base policy operates within its intended performance envelope.

## E.6. Empirical Verification Under Nominal Conditions

We empirically verify nominal performance preservation by evaluating the frozen SAC policy with and without the residual controller enabled in the absence of faults, across multiple random seeds. On `HalfCheetah-v5`, the mean episodic return

under nominal conditions differs by less than $0.5\%$ between SAC-only and SAC with the residual controller enabled, with comparable variance across seeds. This confirms that the residual pathway does not degrade nominal performance.

### E.7. Residual Magnitude and Reward Visualization

For diagnostics, we report the $\ell_2$ norm of the residual action $\|\Delta a_t\|_2$, which provides a compact measure of corrective effort across joints. Reward visualizations use the EMA $\bar{R}_t$ to reduce high-frequency stochasticity. All quantitative results reported in the main paper are computed using undiscounted episodic returns.

As shown in Fig. 2(c) in main paper, when faults are absent or removed, the residual gate collapses, the residual magnitude decays to zero, and performance converges to the nominal baseline. These results demonstrate that the proposed residual controller improves robustness under post-training faults without interfering with nominal behavior.

## F. Policy Consolidation Details

This appendix describes the policy consolidation procedure used in Inference-Time Adaptation and Policy Consolidation section. Consolidation is evaluated as a *post-adaptation* mechanism that transfers stabilized residual corrections into a static policy adapter under sustained faults. It is implemented offline after the residual response stabilizes, rather than as an online automated trigger. Its purpose is not to replace inference-time adaptation, but to characterize a complementary regime in which persistent corrective behavior can be absorbed into fixed parameters, allowing the cerebellar pathway to disengage once compensation has stabilized.

### F.1. Motivation and Scope

The proposed cerebellar residual controller is designed for fast, conservative inference-time recovery. Under mild or transient faults, the residual produces state-dependent corrections that vary over time, making permanent absorption neither necessary nor desirable. However, under sustained faults, inference-time adaptation often converges to a consistent corrective mapping, reflected in the slow residual head.

Policy consolidation leverages this convergence by transferring the stabilized correction into a static adapter. Once consolidated, the adapter reproduces the corrective behavior without online plasticity, effectively freeing the cerebellar pathway from continued intervention. Consolidation is therefore evaluated only as an *offline* mechanism applied after successful inference-time adaptation, and is not used during deployment-time recovery.

Importantly, consolidation is performed *independently for each fault severity and fault type*. We do not attempt joint consolidation across severity sweeps or fault families, as corrective mappings differ substantially across conditions and joint fitting was empirically unstable.

### F.2. Data Collection

For each fault condition, adaptive rollouts are first executed using the full inference-time cerebellar controller. Data collection begins only after an initial transient period, ensuring that the residual correction has stabilized. At each timestep, we record the filtered feature vector $\Phi_t$ and the executed residual action $\tau_t$ produced by the slow residual pathway.

No cerebellar internal states, error signals, phase variables, fault labels, or reward information are used during consolidation. The dataset consists solely of feature–action pairs $(\Phi_t, \tau_t)$ harvested from stabilized adaptive behavior.

### F.3. Adapter Fitting

Consolidation is implemented as a lightweight linear regression that fits a static adapter mapping features to residual actions. Given a dataset $\{(\Phi_i, \tau_i)\}_{i=1}^{N}$, the adapter weights $W_{\text{adapt}}$ are obtained via ridge regression:

$$W_{\text{adapt}} = \arg\min_{W} \sum_{i} \|\tau_i - W\Phi_i\|^2 + \lambda\|W\|^2.$$

This procedure preserves the linear-in-parameters structure of the residual pathway and requires no gradient-based reinforcement learning, no replay buffer, and no critic updates. The resulting adapter produces residual corrections directly from features, without any online learning or meta-adaptation.

**F.4. Evaluation Protocol**

After fitting, the adapter is evaluated on the *same fault condition* from which data were collected. During evaluation, the cerebellar residual controller is disabled entirely, and the executed action is computed as

$$a_t = a_t^{\text{base}} + g\,\tau_t^{\text{adapt}},$$

where $\tau_t^{\text{adapt}} = W_{\text{adapt}}\Phi_t$.

This evaluation isolates the effect of consolidation by measuring performance when corrective behavior is provided solely by the static adapter. No additional learning, parameter updates, or confidence gating are active during this phase.

**F.5. Interpretation**

Consolidation performs robustly across representative actuator, dynamic, and environmental faults. Increased variability is observed for faults that induce global, strongly state-dependent dynamics shifts, where a single static mapping may be insufficient to capture all corrective structure. Nevertheless, for sustained faults where inference-time adaptation converges to a consistent correction, consolidation reliably matches or exceeds online adaptation performance.

These results support a lifecycle interpretation of the proposed architecture: inference-time cerebellar adaptation provides rapid, reversible recovery, while consolidation absorbs stabilized corrections when faults persist, allowing the cerebellar pathway to disengage without sacrificing performance.

# G. Extension to Manipulation: PandaReach-v3

To assess whether inference-time residual adaptation generalizes beyond locomotion, we additionally evaluate the proposed method on `PandaReach-v3`, a goal-conditioned manipulation task with contact-rich dynamics. The task requires controlling the end-effector of a 7-DoF robotic arm to reach a target position within a fixed episode horizon. A SAC policy trained with hindsight experience replay (HER) is used as the frozen base controller.

**Inference-Time Residual Adaptation.**   The cerebellar residual architecture is adapted to manipulation by replacing periodic phase indexing with a time-based task progress variable. Thus, phase is treated as one instance of a more general progress variable rather than a required periodic gait coordinate. Microzones partition the episode horizon into fixed temporal bins, and a dual-head residual structure (fast and slow pathways) is maintained to capture transient and persistent corrective behavior. The residual operates purely at inference time and augments the frozen base policy without retraining or system identification.

The residual input consists of the robot observation, achieved goal, desired goal, and their relative offset. Learning is driven by a goal-directed error signal proportional to the distance and direction to the target, enabling state-dependent corrective actions during execution. This preserves the same progress-conditioned residual-control structure while replacing locomotion-specific phase lookup with a manipulation-specific progress surrogate. A lightweight confidence mechanism modulates residual gain based on task progress and success signals.

**Fault Model and Evaluation Protocol.**   We evaluate post-training actuation faults applied at execution time in the form of uniform actuator scaling, which reduces the magnitude of all control commands by a fixed factor. Faults are applied identically for the frozen baseline and the adaptive controller. Performance is measured using success rate, episodic return, and final distance to the target. Each fault condition is evaluated over multiple random seeds and episodes per seed, and all reported metrics are averaged across rollouts.

**Results.**   Across evaluated fault conditions, inference-time residual adaptation consistently improves success rate and reduces final distance to the goal relative to the frozen SAC baseline under moderate fault severities. Performance degrades gracefully as fault magnitude increases, reflecting the increasing difficulty of compensating large actuation errors without replanning or retraining. These results indicate that conservative inference-time residual adaptation is not specific to locomotion, and extends to manipulation tasks where localized post-training degradation disrupts task execution.

