# OpenReview forum: "Cerebellar-Inspired Residual Control for Fault Recovery: From Inference-Time Adaptation to Structural Consolidation"
_ICML.cc/2026/Conference — ICML 2026 regular_

### Official Review · Reviewer_5fVa · 2026-03-12

**Soundness:** 3
**Presentation:** 2
**Significance:** 2
**Originality:** 2
**Overall Recommendation:** 4
**Confidence:** 3

**Summary:**

This paper addresses the challenge of robot fault recovery (e.g., actuator degradation, dynamics shifts) during deployment, where retraining or exploration is often impractical. The authors propose a cerebellar-inspired residual control framework that operates at inference time. Instead of retraining the base policy or performing explicit system identification, this framework introduces a parallel, adaptive residual path. The proposed architecture incorporates several biologically inspired components, including: High-dimensional feature expansion for pattern separation, Parallel microzone-style residual pathways for localized corrections, Dual-Time Scale Plasticity using excitatory and inhibitory eligibility traces. Structural consolidation mechanism that converts persistent residual corrections into long-term model adjustmentsThe effectiveness of the proposed method is evaluated across multiple MuJoCo environments, considering various post-training faults including actuator faults, dynamic perturbations, and environmental disturbances.

**Compliance With Llm Reviewing Policy:**

Affirmed.

**Final Justification:**

My questions are fully addressed. Therefore I raise my score.

**Key Questions For Authors:**

Key questions:
- Could the authors comment on how well the proposed framework is expected to generalize to more complex robotic control settings, such as high-dimensional manipulation tasks, contact-rich environments, or real-world robotic platforms( humanoid robots walking stairs)?
- Fairness of experimental comparisons. In Figure 3, the robustness under different fault severities is compared primarily against SAC trained in fault-free settings, where performance degradation under faults is expected. Similarly, in Figure 2(c), the comparison mainly involves policies that do not explicitly address fault scenarios. Could the authors provide additional quantitative comparisons with stronger baselines that are specifically designed to handle system disturbances or faults?
-According to the quantitative results presented in Table 1, the proposed method does not demonstrate a decisive advantage over other adaptive baselines. Given the additional architectural complexity (e.g., high-dimensional expansion and dual-trace filtering) introduced by this cerebellar-inspired approach, the lack of substantial performance gain raises questions about whether the increased computational cost is justified by the resulting benefits.

**Limitations:**

yes

**Strengths And Weaknesses:**

Strengths:
- Well-motivated problem: The paper targets an important and practical challenge in robotics: enabling reliable adaptation to unexpected faults during deployment without retraining the policy.
- Interesting biological inspiration: The work introduces several cerebellar-inspired principles, including pattern separation, microzone-like modular corrections, and eligibility-trace-based plasticity, This provides a conceptually appealing combination between computational neuroscience and learning-based control.
Weekness:
- Although cerebellar inspiration is mentioned, the paper provides limited background on the biological concepts (e.g., microzones and their functional roles), which may make it difficult for readers outside neuroscience to fully understand the motivation behind the architectural design.
- While the framework is conceptually sound, the presentation of certain mathematical derivations and notations is somewhat ambiguous. This lack of mathematical detail makes it challenging for readers to fully grasp the implementation nuances without referring to the appendix or code.
- The comparison between the proposed adaptive framework and the static SAC baseline may be inherently biased. The SAC baseline is a frozen, pre-trained policy that has never encountered fault scenarios during its training phase. It is expected that a static policy would underperform when facing out-of-distribution disturbances.
- The experiments are restricted to MuJoCo simulations tasks. It remains unclear how well the proposed framework generalizes to more complex robotic control settings or real-world systems.
- While the method is described as lightweight, components such as high-dimensional feature expansion and parallel trace processing could introduce additional computational overhead. The paper does not provide a detailed analysis of inference time or computational cost.

---

> ### Author Rebuttal · Authors · 2026-03-30
>
> We thank the reviewer for the thoughtful feedback. We clarify below to address points that may not have been sufficiently clear.
>
> $\textbf{On fairness of comparisons:}$ Our evaluation is not primarily based on a frozen SAC policy. The core results (Tables 1–2) compare against disturbance-aware baselines, including Robust SAC, SAC+OSI, SAC+Offline Res., and CMAC, Online LMS, spanning training-time robustness, system ID, and residual adaptation. The SAC baseline in Fig. 3 serves only to illustrate degradation of nominal policies under increasing fault severity, not as the main comparison. Our central claim is that the proposed method remains competitive with these baselines while requiring neither fault exposure during training nor explicit system identification at deployment. We will revise to clarify this.
>
> $\textbf{On additional results across fault severities:}$ We performed full fault severity sweeps for all methods (actuator scaling, bias, mass, damping, friction increase and decrease). Our method maintains higher returns across severities. Under actuator scaling, it retains $\sim$40–60% of nominal performance even at high severity (0.5–0.4), while several baselines collapse to near-zero. Under damping and mass increases, competing methods show sharp mid-range drops, whereas ours degrades more gradually and remains stable. Similar trends hold for actuator bias and friction, where our method avoids abrupt failures seen in baselines. Overall, this shows consistent performance across severities rather than tuning to specific regimes. We will include these results in the revision.
>
> $\textbf{On lack of decisive improvement:}$ We agree our method does not dominate every baseline in every setting; however, this is not the objective. Our contribution is consistent performance across fault families and environments under minimal assumptions. From Tables 1–2, competing methods are strong only in specific regimes (e.g., identifiable dynamics for OSI or particular residual structures), whereas ours ranks first or second across actuator, dynamic, and environmental faults, including Humanoid-v5. Notably, it achieves the best performance across all three fault families on Humanoid-v5 while remaining competitive. This matters because we target unknown post-training faults where structure cannot be assumed; methods relying on anticipated or structured faults address a more restricted setting.
>
> $\textbf{On whether the added structure is justified:}$ The additional structure is not cosmetic, but directly tied to the observed robustness. The controller is
> $
> a_t = \pi_\theta(s_t) + a_t^{\text{res}}, \quad
> a_t^{\text{res}} = g_t \left( W_t^{\text{fast}} + W_t^{\text{slow}} \right)\Phi_t
> $
> where the gain $g_t$ enforces bounded and conservative intervention, and $(W^{\text{fast}}, W^{\text{slow}})$ separate transient from persistent corrections. The feature $\Phi_t$ is constructed from fixed projections with dual temporal traces, enabling localized and time-scale aware adaptation. Ablation results (Table 3) show that removing any of these components leads to consistent performance degradation, indicating gains arise from locality, temporal structure, and gating rather than residual learning alone.
>
> $\textbf{On biological motivation and clarity:}$ We agree that the terminology can be made clearer. The biological framing is not required to understand the method and serves only to motivate the design. In particular, “microzones” correspond to phase-conditioned local residual modules, and “residual authority” corresponds to the scalar gain $g_t$. We will revise the paper to define these concepts explicitly, and reduce reliance on biological terminology for readability.
>
> $\textbf{On generalization beyond MuJoCo:}$ We note that our method is not limited to cyclic locomotion. We already include PandaReach-v3, where the phase mechanism is replaced by task progress while retaining the same residual structure, demonstrating applicability to non-cyclic manipulation. More generally, the framework requires only state measurements and a nominal rollout, and does not depend on simulator-specific assumptions.
>
> $\textbf{On computational overhead:}$ The residual module consists of a fixed feature projection and linear updates with bounded gain. It introduces no gradient-based optimization, replay buffer, or system identification, and operates alongside the base policy. This results in minimal overhead relative to the policy forward pass. We agree that explicitly reporting runtime would strengthen the paper and will include this in revision.
>
> We appreciate the reviewer’s concerns and agree that clarity can be improved. At the same time, we would like to gently mention that the paper already evaluates against disturbance-aware baselines, demonstrates consistent performance across diverse fault regimes, and provides ablations supporting the architectural design. We hope these clarifications address the concerns and better reflect the contribution.

---

> > ### Author Rebuttal · Reviewer_5fVa · 2026-04-03
> >
> > Thank the authors for the response. My questions are addressed. Therefore I will raise my score.

---

### Official Review · Reviewer_uByq · 2026-03-13

**Soundness:** 2
**Presentation:** 2
**Significance:** 2
**Originality:** 2
**Overall Recommendation:** 3
**Confidence:** 4

**Summary:**

The paper proposes an approach to adapting to post-training faults, where retraining, exploration, or system identification are impractical. It introduces an inference-time, cerebellar-inspired residual control framework that augments a frozen reinforcement learning policy with online corrective actions, enabling fault recovery without modifying base policy parameters. Residual corrections are learned from local tracking errors and applied additively, preserving the structure of the learned policy.

**Compliance With Llm Reviewing Policy:**

Affirmed.

**Final Justification:**

The paper is of interest and the authors have clarified a few points. However, there are a few remaining points, namely the limited biological inspiration (or clarification thereof), the very limited discussion of related works, and the limited theoretical claims and formal justification of the approach (e.g. there are no performance guarantees). It would seem to me they are too strong limitations to ignore.

**Key Questions For Authors:**

- Baselines: the baselines are not referenced. It should be made clear how they connect to the literature.

- The biological inspiration. “Residual authority” ? Why not just say the gain of the residual? Is this a common term? If so, please reference.

- the authors explain “... the regulation of corrective authority. Corrective signals
must be bounded, assistive, and suppressible, rather than
globally optimizing behavior. This aligns with the functional
role of the cerebellum in motor control, which operates as
an adaptive corrector layered on top of a stable controller,
with fast error-driven correction and slow consolidation of
persistent corrections into long-term structure.” Where is the referencer for this role of the cerebellum and what evidence is it based on?

- In the related work, the authors cite two studies (Albus, 1971; Kawato & Gomi, 1992)

- The authors refer a lot to microzones. The authors should more clearly and early explain cerebellar microzones and other bio-inspired concepts and how these relates to the proposed architecture.

- “To enable localized, context-dependent adaptation, the residual pathway is partitioned into phase-conditioned micro-zones.” I don’t think most people will be able to understand this since the meaning and utility of micro-zones has not been explained.

- In appendix A.3: “Microzone outputs are blended using smooth phase-dependent weights wk (ϕt ).” Blended how?

- The adaptation/error correction procedure. Could you explain the procedure for adaptation? How does the controller know the performance has declined and what kind of information is being fed to train the residual controller? It is mentioned that around equation 17 that a sustained elevation of the residual energy indicates the need to adapt. But that residual energy seems already to be based on the residual action, which should still be learned? The protocol is unclear to me.


- Stability properties. The only result seems to be that the inference-time adaptation cannot inject unbounded control inputs. Putting a range on a network is not really a challenging problem to be solved.

- Eq. 16: do you mean lambda is in [0,1]? and $g_t \to 0$ as $t \to \infty$?

- what is the purpose of estimating the phase instead of time itself? And what makes the data periodic?

- Theory. Is there a fundamental reason why a residual controller would work better than alternatives? What kind of theoretical assumptions does this make? Does your technique provide any guarantees? Are those guarantees improved over related techniques.

**Limitations:**

Yes

**Strengths And Weaknesses:**

Strengths:
- Empirically good performance on a variety of faults
- The technique makes sense inuitively as it allows a decoupling between adaptation and the controller from training.

Weaknesses:
- the experimental protocol for adaptation is not clearly explained
- the biological inspiration is not clearly explained
- the theoretical properties and key benefits are poorly understood or not explained
- the connection to other residual controller techniques is not so clearly described
- the baselines are not explained carefully in relation to prior works

---

> ### Author Rebuttal · Authors · 2026-03-30
>
> We thank the reviewer for the detailed feedback. We agree several aspects can be clarified, and we address each of the raised questions explicitly below.
>
> $\textbf{On baselines and their connection to prior work:}$ The baselines correspond to standard approaches for handling post-training disturbances. Robust SAC represents training-time robustness via domain randomization. SAC+OSI corresponds to dynamics-conditioned policies with online system identification. SAC+Offline Res. represents residual policy learning with offline-trained corrective actions. CMAC and Online LMS are classical and modern forms of inference-time residual adaptation based on local error signals. We agree that these connections were not stated clearly enough and will revise the related work and experimental sections to explicitly link each baseline to its corresponding literature.
>
> $\textbf{On biological inspiration and terminology:}$ The term residual authority corresponds to the scalar gain $g_t$ that scales the residual correction, and we will replace it with “residual gain” for clarity. Our use of cerebellar terminology is motivated by feedback error learning models (Albus, 1971; Kawato & Gomi, 1992), where the cerebellum acts as an adaptive feedforward corrector over a stable controller. In our method, this corresponds to an additive residual controller with bounded gain and local error-driven updates.
>
> $\textbf{On microzones and their implementation:}$ Microzones correspond to phase-conditioned local residual modules that restrict adaptation to regions of the trajectory. At time $t$, the phase $\phi_t$ activates a small subset of modules via smooth weights $w_k(\phi_t)$ satisfying
> $\sum_{k} w_k(\phi_t) = 1, \quad w_k(\phi_t) \ge 0$
>
> The residual is computed as
> $a_{\text{res}} = g_t \sum_{k} w_k(\phi_t) \left( W_k^{\text{fast}} + W_k^{\text{slow}} \right) \Phi_t$
> This blending corresponds to a convex combination of local residual outputs. We will move this explanation to the main text.
>
> $\textbf{On the adaptation and error-correction procedure:}$ Adaptation is driven by local tracking error relative to a reference trajectory. Specifically,
>
> $e_t = q_d(\phi_t) - q_t, \quad
> \dot{e}_t = \dot{q}_d(\phi_t) - \dot{q}_t, \quad
> r_t = \dot{e}_t + \Lambda e_t$
>
> The residual weights are updated using
>
> $\Delta W_t = \eta \frac{r_t\Phi_t^\top}{\|\Phi_t\|_2+\epsilon}$
>
> Activation of the residual is not based on residual energy. Instead, it is governed by a performance-based meta-controller:
> $\bar R_t = (1-\rho)\bar R_{t-1} + \rho R_t , \quad g_{t+1} = g_t + \kappa\,\mathbb{I}[\bar R_t < \bar R^\star - \delta] - \lambda g_t$
>
> Thus, sustained degradation in reward activates bounded residual correction, while local error drives learning. Residual energy is used only to detect persistent mismatch for consolidation, not to initiate adaptation.
>
> $\textbf{On stability properties and guarantees:}$ We agree that this should be clarified. The goal is not global optimality but safe, bounded adaptation. The method enforces
> $\| a_{\text{res}} \| \le g_{\max} W_{\max} \Phi_{\max}$,
> ensuring bounded control input. In addition, under nominal conditions,
> $g_t \to 0 \;\Rightarrow\; a_{\text{res}} \to 0$,
> which guarantees non-interference with the base policy. These properties differ from classical adaptive control, where guarantees rely on structured uncertainty or known models. Our method instead provides bounded, conservative adaptation under unknown faults. We will clarify this distinction in the main paper.
>
> $\textbf{On Eq. 16:}$ In Eq. 16, $\lambda$ denotes the decay rate governing the dissipative dynamics of the residual gain. Under nominal conditions, this ensures $g_t \to 0$, causing residual actions to vanish and recovering the base policy behavior. This formalizes the non-interference property.
>
> $\textbf{On phase estimation and periodicity:}$ Phase $\phi_t$ is used instead of time to align trajectories under varying execution speeds in cyclic tasks. Time-based indexing would misalign reference trajectories under disturbances. Phase is estimated from joint kinematics, capturing progression along the motion cycle. For non-periodic tasks, phase is replaced by task progress, as demonstrated in PandaReach. Thus, periodicity is not assumed globally but used where appropriate.
>
> $\textbf{On theoretical justification:}$ The advantage of residual control arises from decomposing control as
> $a_t = \pi_{\theta}(s_t) + a_t^{\mathrm{res}}​$,
> which separates long-horizon behavior (handled by $\pi_\theta$) from fast disturbance rejection (handled by $a_t^{\mathrm{res}}$). The bounded gain $g_t$ and local updates prevent instability and drift, which are common in naive residual adaptation. The benefit is therefore not from residual learning alone, but from its regulation through locality, boundedness, and performance-triggered activation.
>
> We appreciate the reviewer’s concerns and agree that clarity can be significantly improved. We will revise the paper to clarify.

---

> > ### Author Rebuttal · Reviewer_uByq · 2026-04-06
> >
> > The authors have clarified a few points but I still have the concern that the theoretical claims are limited.There are many update rules and short motivations for each such rule, but not really a comprehensive guarantee of why, when, how, etc. the algorithm as a whole would work.

---

> > > ### Author Response · Authors · 2026-04-07
> > >
> > > We thank the reviewer for the follow-up. We clarify here the unified principle underlying the algorithm and how the full system operates as a whole.
> > >
> > > Rather than viewing the method as a collection of separate update rules, it is more naturally understood as a *two-level closed-loop adaptive system* consisting of:\
> > > (i) an inner loop that performs local, error-driven residual adaptation, and \
> > > (ii) an outer loop (meta-controller) that regulates how much residual correction is applied based on sustained performance trends.
> > >
> > > We present this in simplified form as
> > > $
> > > a_t = \pi_\theta(s_t) + g_t \ W \Phi_t
> > > $
> > > where in the full model the residual is phase-conditioned with microzone-specific fast and slow weights (Eq.\~6). This view helps explain why, when, and how the method works.
> > >
> > > $\textbf{Why it works:}$
> > > The residual learns a bounded correction to the mismatch between nominal and faulted dynamics via a normalized, regularized update (Eq.\~9), implemented with L2 regularization (Table\~4, Appendix\~A.4), bounded features, and deadzones.
> > > Under stationary fault conditions, this behaves like a regularized LMS update and settles to a stable correction, rather than drifting as in unconstrained residual learning.
> > >
> > > $\textbf{When it acts:}$
> > > Residual adaptation is governed by a performance-driven meta-controller. So the residual is only activated when there is sustained performance degradation. When performance is nominal, Eq.\~16 enforces dissipative gain dynamics $g_{t+1} = (1-\lambda)g_t$, implying $g_t \to 0$, so $a_t^{res} \to 0$ and the controller reduces to the base policy.
> > > In practice, this is further reinforced by the deadzone (which suppresses small-error updates) and decay in the fast head, so residual activity naturally disappears when it is not needed.
> > >
> > >
> > > $\textbf{How the full system behaves:}$
> > > Taken together, the method forms a bounded closed-loop system.
> > > The inner loop adapts residual weights to reduce tracking error, while the outer loop regulates the strength of this adaptation.
> > > Because the residual output is bounded (via clipping and constrained weights), the gain $g_t$ is bounded, and a directional-consistency gate suppresses corrections that oppose the base policy, the total control input remains bounded for all time, independent of the fault.
> > >
> > > $\textbf{A critical question discussed by this paper}$ is whether recovery can be achieved without system identification, retraining, or exploration at deployment.
> > > Our results indicate that this is possible when adaptation is structured as *bounded, performance-driven local correction*, rather than unconstrained learning, where updates can grow unchecked and interfere with the base policy.
> > >
> > > $\textbf{Scope of guarantees:}$
> > > We do not claim global optimality or guarantees under arbitrary non-stationary faults.
> > > Instead, the method ensures,\
> > > (i) bounded control inputs, \
> > > (ii) non-interference under nominal conditions via dissipative gain dynamics, and \
> > > (iii) bounded and empirically convergent weight adaptation under stationary disturbances, consistent with standard regularized LMS behavior.
> > >
> > > Notably, competing inference-time baselines (e.g., SAC+CMAC, SAC+Online LMS) operate under similar assumptions in this setting and do not provide stronger composite guarantees, while also lacking performance-driven regulation and timescale separation.

---

### Official Review · Reviewer_wgSU · 2026-03-20

**Soundness:** 4
**Presentation:** 4
**Significance:** 4
**Originality:** 3
**Overall Recommendation:** 5
**Confidence:** 4

**Summary:**

Motivated by this cerebellar correction mechanism, the authors propose an inference-time residual control architecture that augments a frozen reinforcement learning policy with lightweight online correction. The proposed residual controller is inspired by traditional adaptive control with two caveats: the controller operates without explicit models or policy gradient updates, and it regulates residual plasticity and authority in a conservative manner using sustained performance signals, thus preserving the base policy's behaviour. Experiments support the efficacy of the residual controller.

**Compliance With Llm Reviewing Policy:**

Affirmed.

**Key Questions For Authors:**

- Can the adaptive controller be applied outside locomotion tasks? Have the authors thought about additional environments and settings? The adaptive controller is well motivated by the cerebellum and appears to be primarily intended for robotic applications, but many policies outside robotics are subject to out of training distribution test-time faults.

**Limitations:**

Yes.

**Strengths And Weaknesses:**

Strengths:
- The cerebellum as a motivating biological structure is interesting.
- The method mitigates out of training-sample test-time faults. That is, the RL policy augmented with the residual controller does not need to be trained in a robust manner, and moreover, not all fault-types need be specified at training.
- The paper is well written and provides a good introduction to related works, well positioning the contributions of the authors.
- Safety and stability analysis is a necessary and positive component of this paper, in particular, for the goal for mitigating (unforeseen) test time faults.
- Controlled adaptation is a neat feature for when the training regime systematically differs from the test regime, or in other words, this enables continual learning.
- Experiments are comprehensive and clearly demonstrate that proposed residual controller is competitive (either attaining the best or second-best performance) amongst competitors. Notably, these experiments include both residual and non-residual competitors, demonstrating that the authors' method is competitive against other test-time fault correction methods and training with faults.

Weaknesses:
- While the exposition is generally clear, it would be useful to provide pseudocode that pieces Section 3 into a single block of pseudocode. Each of the components are well described, but presenting the complete algorithm may be useful for readers not familiar with this literature.

---

> ### Author Rebuttal · Authors · 2026-03-30
>
> We thank the reviewer for the positive assessment and the constructive suggestions.
>
> $\textbf{On applicability beyond locomotion}$ Yes, the proposed adaptive residual controller is not specific to locomotion. The core mechanism is task-agnostic: augmenting a frozen policy with a lightweight, bounded residual driven by local error signals and modulated by performance feedback. This requires only (i) a nominal policy, (ii) a suitable error or deviation signal from observations, and (iii) additive corrective actions.
> What is locomotion-specific in our current implementation is the use of a phase variable for indexing, which leverages periodicity. More generally, the framework is indexing-agnostic and can incorporate alternative progress variables (e.g., time, trajectory progression, or task-specific state abstractions), as demonstrated in our PandaReach experiments.
> A key assumption is that the base policy remains approximately valid under the fault, such that corrections are local and bounded. In cases of catastrophic failure, the method is designed to abstain rather than override the base policy.
>
> $\textbf{On presentation clarity}$ We appreciate the suggestion regarding pseudocode. We will improve the presentation by providing a more consolidated and structured description of the full algorithmic pipeline (currently distributed across Section 3) to make implementation clearer for readers.

---

> > ### Author Rebuttal · Reviewer_wgSU · 2026-04-05
> >
> > The authors have answered question regarding the applicability of the adaptive controller beyond locomotion. The authors have also agreed to include a consolidated presentation of their algorithm.

---

### Official Review · Reviewer_DKSQ · 2026-03-23

**Soundness:** 3
**Presentation:** 2
**Significance:** 3
**Originality:** 2
**Overall Recommendation:** 5
**Confidence:** 3

**Summary:**

This paper presents a cerebellar-inspired residual controller designed for post-training fault recovery in reinforcement learning (RL)-based control systems. The core idea is to augment a frozen base policy with an online adaptive residual pathway that remains inactive under nominal conditions and provides bounded corrective actions only when performance degradation is detected.

**Compliance With Llm Reviewing Policy:**

Affirmed.

**Final Justification:**

My questions are fully addressed. Therefore I raise my score.

**Key Questions For Authors:**

* Can you provide a more precise description of the phase-portrait estimator? Which joint(s) are used for estimation in each environment? How is the mapping from joint angles and velocities implemented in practice? This is essential for reproducing your results.

* In Humanoid experiments, faults were applied only to a subset of lower-body joints. Did you experiment with applying faults to all joints? If so, what were the results? If not, do you have any concerns about how the method would perform under more comprehensive fault patterns?

*The phase-based reference indexing appears central to the method’s success in locomotion. For non-periodic tasks like manipulation, you used time-based segmentation instead. Do you view this as a fundamental limitation of the approach? Could phase estimation be extended to non-periodic tasks using alternative progress measures (e.g., task completion ratio, end-effector trajectory progress)?

* The policy consolidation results show that a static linear adapter can replace the adaptive residual under sustained faults. How do you determine when consolidation should be triggered in practice? Is this an offline process performed after deployment, or could it be automated online?

* The meta-controller modulates residual authority based on reward thresholds and stagnation horizons. How sensitive is performance to these parameters across different environments and fault types? Did you observe any regimes where improper tuning led to interference or instability?

**Limitations:**

Yes

**Strengths And Weaknesses:**

## Strengths ##
* The paper addresses a realistic and important challenge—deployment-time recovery under unknown, persistent faults without retraining. This setting has clear practical relevance for real-world robotic systems where post-deployment adaptation is critical.

* The cerebellar-inspired design (microzones, dual-timescale learning, performance-driven gating) is not merely a metaphor; it translates into concrete mechanisms that are shown to work across multiple environments and fault types. The ablation studies reasonably support the contribution of individual components.

* The evaluation spans three locomotion environments with increasing complexity, three distinct fault families across multiple severity levels, and one manipulation task. Comparisons include multiple strong baselines (e.g., SAC+OSI, Robust SAC, CMAC, online LMS), and results are reported with proper statistical measures.

* The paper explicitly addresses the “do-no-harm” property—residual activation under nominal conditions is suppressed, and bounded authority is enforced. This is critical for deployment scenarios.

* The appendix provides extensive implementation details, hyperparameter tables, additional ablation results, and protocol descriptions, which significantly support reproducibility.

## Weakness ##

* The phase-portrait estimator is a key component for phase-indexed reference lookup, yet its implementation is only vaguely described as “deterministic” and “nearest-neighbor lookup.” No concrete algorithm, equations, or chosen joint variables are provided, making reproduction difficult.

* For the Humanoid-v5 task, faults are applied only to a subset of lower-body joints (hips, knees, ankles) rather than all actuated joints. While the authors justify this to avoid immediate collapse, it reduces the severity of the evaluation and leaves open the question of how the method would perform under more distributed fault patterns.

* The paper is dense with repetitive content (e.g., the episodic residual energy definition appears multiple times). Key conceptual explanations are often interleaved with implementation details, and the separation between main text and appendix could be more thoughtfully structured. This makes the paper unnecessarily tiring to read.

* The paper deliberately avoids convergence or optimality guarantees, offering only basic boundedness and non-interference properties. While this is understandable given the problem complexity, it limits the depth of the contribution compared to work that provides stronger stability arguments.

* Although the method is evaluated on PandaReach (a manipulation task), the phase mechanism is replaced by time-based task progress, which is a significant departure. The results on this task are presented briefly in the appendix, and the effectiveness of the core “phase” concept is not convincingly extended beyond locomotion.

---

> ### Author Rebuttal · Authors · 2026-03-30
>
> We thank the reviewer for the detailed and constructive questions. We clarify each point below.
>
> $\textbf{On the phase estimator and implementation details:}$
> The phase estimator is a deterministic phase-portrait mapping computed from a single joint coordinate. Given a selected joint position $q_i$ and velocity $\dot{q}_i$, we compute
>
> $phase = \frac{\mathrm{atan2}(\beta \dot{q}_i,\ q_i - \bar{q}_i) + \pi}{2\pi} \in [0,1)$
>
> where $\bar{q}_i$ is an exponential moving average of $q_i$, and $\beta$ scales velocity to match angular magnitude. The resulting phase is further smoothed using a circular exponential moving average to ensure temporal continuity.
>
> In all environments, the estimator uses a single joint coordinate:
>
> HalfCheetah: $q[0], qd[0]$, where $q = obs[2{:}8]$, $qd = obs[11{:}17]$ \
> Ant: $q[4], qd[4]$, where $q = obs[6{:}14]$, $qd = obs[14{:}22]$ \
> Humanoid: $q[5], qd[28]$, where $q = obs[5{:}22]$, $qd = obs[28{:}45]$
>
> We selected joints exhibiting the most stable periodic behavior. A single dominant joint was sufficient; additional joints did not improve results.
>
> The mapping is implemented as:
>
> (1) compute phase from $(q_i, \dot{q}_i)$.\
> (2) compute circular distance to reference phases.\
> (3) retrieve $(q_d, \dot{q}_d, \ddot{q}_d)$ via lookup in phase space.
>
> $\textbf{On humanoid faults across all joints:}$
> We did experiment with applying faults to all joints. In this setting, the humanoid typically collapses immediately across methods, resulting in near-zero rewards and preventing meaningful evaluation of recovery dynamics. This reflects a known stability limit in RL locomotion, where large perturbations cause immediate collapse.
> Our goal is to study recoverable but challenging perturbations, where adaptation behavior can be meaningfully observed and compared. We therefore focus on lower-body joints (hips and proximal leg joints), which dominate locomotion stability and produce severe but still recoverable degradation.
> Importantly, the method itself is not restricted to partial faults, the residual operates over the full action space. However, full-body faults correspond to a substantially harder regime involving global coordination failure rather than localized post-training degradation. We will clarify this evaluation choice.
>
> $\textbf{On phase indexing and non-periodic tasks:}$
> The framework is indexing-agnostic: phase is one instance of a low-dimensional progress variable used for locomotion. For locomotion, phase provides a natural progress variable. For non-periodic tasks (e.g., manipulation), we instead use task progress surrogates such as time or trajectory progression. The core mechanism of conditioning residual adaptation on a low-dimensional progress variable remains unchanged.
> We therefore do not view phase as a fundamental limitation of the approach. More generally, the same mechanism can incorporate alternative progress measures (e.g., task completion ratio or end-effector trajectory progress) enabling extension beyond periodic locomotion.
>
> $\textbf{On the consolidation mechanism:}$
> In the current work, consolidation is implemented as an offline, post-adaptation procedure. For a given fault condition, we follow a collect–fit–evaluate pipeline\
> (1) collect feature–action pairs from the slow residual pathway after an initial transient.\
> (2) fit a static linear adapter via ridge regression.\
> (3) evaluate by disabling the adaptive residual and deploying the learned adapter.
>
> The trigger is not automated, consolidation is applied once the residual stabilizes under a sustained fault. While the formulation supports automated triggering, we leave this for future work. Conceptually, consolidation reflects a lifecycle: fast adaptation enables recovery, while consolidation absorbs persistent corrections and frees adaptive capacity.
>
> $\textbf{On meta-controller sensitivity:}$
> The meta-controller parameters are bound and updated smoothly based on reward EMA signals. We include a sensitivity study in the appendix (varying EMA factor, reward-drop threshold, and stagnation horizon), which shows that performance changes moderately but preserves the same qualitative trends across fault severities.
> Empirically, we did not observe catastrophic instability. Two failure modes were observed during tuning: (i) overly tight reward-drop thresholds can cause premature gain escalation and oscillatory corrections, and (ii) very short stagnation horizons can trigger adaptation before the slow head stabilizes. These effects are mitigated by residual clipping, deadzones, directional gating, and bounded parameter updates. We will make these findings more explicit in the main text.
>
> This design emphasizes bounded, conservative adaptation rather than convergence or optimality guarantees under unknown faults.
>
> $\textbf{On presentation clarity:}$
> We will streamline the manuscript by removing redundancies and improving the separation between conceptual description and implementation details.

---

> > ### Author Rebuttal · Reviewer_DKSQ · 2026-04-02
> >
> > Thank the authors for the response. My questions are fully addressed. I will raise my score to 5.

---

### Decision · Program_Chairs · 2026-04-30

**Decision:**

Accept (regular)

**Comment:**

This paper introduces a cerebellar-inspired residual controller for inference-time fault recovery in RL-based robotic control. A frozen base policy is augmented with a lightweight adaptive residual that kicks in only when sustained performance degradation is detected, applying bounded corrective actions without touching the base policy. The design draws on cerebellar principles—phase-conditioned microzones, dual-timescale weight updates, and a conservative meta-controller—and is evaluated across MuJoCo locomotion tasks and PandaReach under actuator, dynamic, and environmental faults.

Four reviews came in at 5, 5, 4, and 3. Three reviewers recommended acceptance; the one weak reject (uByq) appreciated the empirical results but wanted stronger theoretical grounding. The rebuttal addressed most concerns well: DKSQ's questions about the phase estimator were fully resolved with concrete equations and joint-index details (score raised to 5); wgSU's question on applicability beyond locomotion was answered via the PandaReach experiments (fully resolved); 5fVa's concerns about baseline fairness and missing runtime analysis were addressed by pointing to the disturbance-aware baselines and committing to add profiling (score raised to 4). uByq's concern about theory was partially addressed—the authors gave a clean two-level closed-loop interpretation and spelled out the bounded-input and non-interference properties—but the reviewer still felt the guarantees were thin. That's a fair point, but the lack of formal convergence guarantees is a reasonable limitation for this problem setting (unknown post-training faults, no model access), and is not unusual in the empirical RL literature.

Overall a reasonable paper on a practically important problem, with a well-designed method, comprehensive experiments, and a rebuttal that resolved the main concerns. I recommend acceptance.